# Unravelling the therapeutic potential of dual TGFβ-1 and CXCR4 inhibition in breast cancer using computational strategies

Burhan Ul Haq[1,☯], Shazia Sofi[1,☯], Hina Qayoom[1], Nusrat Jan[1], Asma Jan[1], Mohammad Aljasir[2], Irshad Ahmad[3], Abdullah Almilabairy[4], Fuzail Ahmad[5], Bader Alshehri[6], Manzoor Ahmad Mir[1]*

1 Cancer Biology Laboratory, Bioresources Department, SBS, Kashmir University, Srinagar, India, 2 Department of Medical Laboratories, Qassim University, Buraidah, Kingdom of Saudi Arabia, 3 Department of Medical Rehabilitation Sciences, King Khalid University, Abha, Kingdom of Saudi Arabia, 4 Department of Family and Community Medicine, Faculty of Medicine, Al Baha University, Al Baha, Kingdom of Saudi Arabia, 5 Respiratory Care Department, College of Applied Sciences, Almaarefa University, Diriya, Saudi Arabia, 6 Department of Medical Laboratory Sciences, College of Applied Medical Sciences, Majmaah University, Majmaah, Saudi Arabia

☯ These authors contributed equally to this work.
* drmanzoor@kashmiruniversity.ac.in

## Abstract

### Background and rationale

The incidence and mortality of breast cancer (BC) continue to increase, making it a matter of public health concern worldwide. Despite the various strides made in breast cancer (BC) research, the molecular mechanisms driving its progression remain incompletely understood, particularly the role of key regulatory genes in tumor development and therapy resistance. TGFβ-1, IL19, CXCR4, BMP1, VCAN, and WNT2 have been implicated to be instrumental to critical oncogenic pathways; however, their cumulative contribution toward the pathophysiology of BC has not yet been investigated. Therefore, the present study utilizes an integrative bioinformatics approach to decipher their functional relevance, providing a basis for targeted therapies.

### Methods and analytical approach

TGF-β1, IL19, CXCR4, BMP1, VCAN, and WNT2 are among the important genes in BC that we have studied in great detail using bioinformatics techniques that detail differential gene expression analysis for their dysregulation in BC. Their broader oncogenic implications were then clarified by performing pan-cancer and pathway enrichment analyses. Molecular docking studies were employed to comprehend the functional interactions and potential therapeutic targets in protein-protein interaction (PPI) networks.

**Data availability statement:** All relevant data are within the paper and its Supporting Information files.

**Funding:** This study was financially supported by the Anusadhan National Research Foundation-ANRF (SERB), Department of Science and Technology, Government of India, in the form of a grant (SERB/CRG-2023/008460) received by MAM. The author Bader Alshehri extends the appreciation to the Deanship of Postgraduate Studies and Scientific Research at Majmaah University for funding this research work through the project number (R-2025-1986). This study was also financialy supported by the deanship of research and graduat studies at King Khalid University in the form of a large Research Project Grant (RCP-2/205/45). The author Shazia Sofi is a senior research fellowship (SRF) recipient from Council of Scientific and Industrial Research (CSIR), Government of India (112-2511-4537/2K23/1).

**Competing interests:** The authors have declared that no competing interests exist.

**Abbreviations:** TGFβ-1, transforming growth factor beta-1; CXCR4, C-X-C motif chemokine receptor 4; IL19, interleukin 19; BMP1, bone morphogenetic protein 1; VCAN, versican; WNT2, wingless-type MMTV integration site family member 2; MM-GBSA, molecular mechanics generalized born surface area; MDS, molecular dynamics simulation; DFT, density functional theory; MEP, molecular electrostatic potential; ADMET, absorption; distribution; metabolism; excretion; and toxicity; GO, gene ontology; TNBC, triple-negative breast cancer; BRCA1/2, breast cancer gene 1/2; HNSC, head and neck squamous cell cancer; UCEC, uterine corpus endometrial carcinoma; CESC, cervical and endocervical cancer; CHOL, cholangiocarcinoma; COAD, colon adenocarcinomas; ESCA, oesophageal carcinoma; LUAD, lung adenocarcinomas; MDM2, murine double minute 2; BLCA, bladder urothelial carcinoma; GDC, genomic data commons; LIHC, liver hepatocellular carcinoma.

## Key findings

It is demonstrated by our study that TGFβ-1 and CXCR4 are critical factors in the tumorigenesis of BC and their inhibition shows interference with tumor-associated pathways in a synergistic way. Computational modelling suggests that concomitant inhibition of these two targets, with D4476 and AMD3100, may show therapeutic value through modulation of certain key signalling cascades.

## Conclusion and significance

This study provide new insights into the molecular basis of BC and support the idea of targeting TGFβ-1 and CXCR4 together for therapy. The above findings lay the foundation for future in vitro and in vivo experiments aimed at demonstrating that inhibition of both factors would be a viable strategy to improve the therapeutic outcome in BC.

## 1. Introduction

Breast cancer's rising prevalence and death rate make it a very serious health issue. BC has the second highest incidence of all malignancies. According to GLOBOCAN 2022 data, BC is the primary cause of cancer-related mortality in women globally, accounting for around 685,000 deaths [1]. Although awareness of breast cancer (BRC) and the variety of available treatment options have significantly improved over the past two decades, both its incidence and mortality rates continue to rise considerably. Triple Negative Breast Cancer (TNBC) does not express ER, PR, or HER2, thus hormone treatment is not an effective method to target the same. The TNBC has a terrible prognosis and is highly aggressive malignancy [2]. Among its four molecular subtypes, TNBC and HER2-positive types are the most aggressive, with high invasiveness, poor prognosis, and significant mortality—TNBC alone accounting for 83% of breast cancer deaths [3]. Early and effective diagnostic methods are crucial for improving prognosis and guiding treatment. Compared to patients without TNBC, those with TNBC have been shown to have a lower survival rate and an earlier risk of relapse, and the BC stem cells (BCSCs), inside the tumor that possess the ability to initiate tumors, are thought to be responsible for this relapse [4]. Over the years, extensive research has been conducted with a focus on the immunological systems that modulate the progression of cancer, which is an appealing area of study [5]. It has been discovered that the cytokine family known as chemokines, which is released by leukocytes, immune cells, tumor cells, and other cell types, controls inflammation and immunological responses [6]. Chemokines are divided into subsets that are associated with inflammation and homeostasis based on their roles and patterns of expression. Typically, inflammation triggers the production of inflammatory chemokines. Leukocytes and other cell types express them, which facilitates the recruitment of inflammatory leukocytes to the injured tissues [7,8]. The complex process of metastasis initiates with the EMT-mediated production of CSCs that

aggressively adhere to the substratum and break down the extracellular matrix (ECM) [9]. Next, these cells intravasate into blood or lymphatic vessels, where they proliferate by avoiding apoptosis, eluding the immune system, and invading distant organs [10,11]. After these CSCs arrive at the secondary site, a process known as the mesenchymal-epithelial transition (MET) causes them to revert to the phenotype of epithelial cells, and they colonize to form secondary tumors [12]. Cancer represents a multifaceted illness majorly defined by the unchecked division and proliferation of cancerous cells that carry genetic or epigenetic alterations [13,14]. By avoiding the human immune reaction, these cells can spread to other distant organs and lead to further cancers through a process termed metastasis [15].

Several factors, including the initial tumor's anatomy, physiology, and molecular features, influence the metastatic dissemination of cancer cells [16]. The malignant cells' metastasis to a certain organ is determined by specific molecular mediators' expression at both the primary and metastatic sites. Be it in the liver, lung, or bone the expression of the chemokine receptor CXCR4 promotes the spread of primary breast cancer cells to the secondary location and research has demonstrated that its silence inhibits metastasis [17]. According to reports, TGF-β and PDGF expression are highly correlated with the facilitation of breast cancer metastasis to the lung and bone [18]. The transforming growth factor-β (TGF-β) signalling pathway plays a crucial role in regulating key cellular processes such as proliferation, differentiation, and immune responses [19]. Disruptions in this pathway are linked to cancer development, contributing to tumor initiation and metastasis. The most powerful cytokine, interleukin-1 (IL-1), is a member of a broad protein family that includes eleven distinct cytokines. Among these, the development of bone malignancies and breast cancer metastases into bone have been linked to IL-1β expression and relapse [20]. The function of the bone morphogenetic protein (BMP3), also referred to as osteogenin, in osteogenesis and bone turnover has long been recognized [21]. Genetic abnormalities in BMP3 have been connected to the formation and development of cancers [22]. In the course of BMP-mediated carcinogenesis, it triggers the smad-dependent pathway [23]. In addition to BMP3, BMP2 and BMP4 have also been shown to trigger invasion, metastasis, and tumorigenesis [24]. TAMs release C-X-C motif chemokine ligand 1 (CXCL1), which is involved in the activation of epithelial-mesenchymal transition (EMT) and promotes the migration and invasion of aggressive breast cancer cells [25]. It binds to its corresponding receptor, CXCR2, and the CXCL1/CXCR2 signalling pathway enhances the capability of cancerous cells to migrate and invade [26]. The CXC chemokine receptor 4, or CXCL12/CXCR4 axis, has been shown to contribute to the growth of primary breast carcinogenesis and is essential for the metastasis of breast cancer cells, which in turn leads to the establishment of secondary tumors [27]. According to Xu et al., breast cancer cells' increased expression of CXCR4 caused them to spread to the liver and lungs [27]. When breast cancer cells express their cognate receptor, CXCR4, CXCL12 functions as a chemoattractant at secondary tumor locations, causing the cancer cells to migrate and homing [28]. Downstream activation of signalling pathways like PI3K, MAPK, and focal adhesion kinase by the CXCR4/CXCL12 axis eventually improves the ability to migrate and invade [29]. A significant component of ECM is versican (VCAN), which binds to hyaluronic acid. VCAN is essential for tissue morphogenesis, encourages the growth of tumors, and improves metastasis [30]. Poor patient survival and prognosis were associated with elevated levels of VCAN in the majority of malignancies, including BC [31]. It typically facilitates the lung metastasis of aggressive breast tumors and acts as a metastatic biomarker [32]. In TNBC patients who have an increased risk of lung metastasis, Wnt/β-catenin signalling displays a part to play in EMT and breast cancer metastasis. Attenuation of/β-catenin transcriptional activity abolished metastasis-associated behaviours in TNBC cells, according to genetic, pharmacological, and functional studies [33]. The upregulation of VCAN is closely correlated with promoter methylation and the clinical features of breast cancer patients. Studies suggest that the breast cancer patient's survival rate gets lower when the VCAN expression level gets higher [34].

The treatment regimen of breast cancer currently involves heavy reliance on the molecular subtype. For example: endocrine therapy for hormone receptor-positive subtypes (e.g., tamoxifen, aromatase inhibitors); HER2-targeted therapy for HER2-positive cancers (e.g., trastuzumab, pertuzumab); and immune checkpoint inhibitors (e.g., atezolizumab) have recently been shown to have some efficacy in certain TNBC subtypes [5]. However, the response of most triple negative

and drug resistant tumors is usually poor with these modalities [2]. Hence, targeting non-redundant signalling axes like TGFβ and CXCR4 is necessitated as they affect progression of metastasis, facilitate epithelial-mesenchymal transition and are implicated in maintenance of cancer stem cells. Potentially, these targets may elicit synergism with existing therapies that are less poised to succeed. Therefore, our study aims to bridge this gap by systematically investigating the prognostic and functional relevance of critical mediators—TGFβ-1, IL19, CXCR4, BMP1, IL1A, VCAN, PDK1, and WNT2—in breast cancer progression and metastasis, and evaluating the potential of their targeted inhibition as a novel combinatorial therapeutic strategy.

In this study, we have highlighted the significance of TGFβ-1, IL19, CXCR4, BMP1, IL1A, VCAN, PDK1, and WNT2 in breast cancer pathology. We sought to elucidate through an *in silico* mechanistic approach the involvement of the relevant genes coupled with their targeting aspect to better comprehend their involvement in the pathophysiology of breast cancer. In this regard, we have designed our study to work upon the expression analysis of the above-mentioned significant prognostic factors, and their specific role in pan-cancer including breast cancer. We also investigated their correlation with breast cancer metastasis, their expression in various cancers and their enrichment with the most significant pathways. The study further elaborated the *in silico* targeting of significant genes from the investigated genes and also the simultaneous blockade of the TGFβ-1 and CXCR4 using their respective inhibitors (D4476 and AMD3100) in single as well as synergistic dimension.

## 2. Materials and methods

### 2.1. UALCAN

UALCAN (http://ualcan.path.uab.edu/) is an online interactive portal to analyze cancer omics data [35]. The goal of this tool is to make cancer OMICS data easily accessible to everyone so that they can identify relevant biomarkers or do in silico validation of the gene(s) of interest. These databases offer information about patient survival as well as graphical representations and graphs displaying expression characteristics. Using the TCGA breast invasive cancer dataset, this study explicitly examined the expression of TGFβ-1, IL19, CXCR4, BMP1, VCAN, and WNT2 throughout PAN cancer, different sample types, age demographics, tumour stages, and patient subclasses.

### 2.2. TIMER

TIMER 2.0 (http://timer.comp-genomics.org/) is an online program employed to comprehensively investigate the expression of genes across cancers. Researchers can examine the relationship between immune cell types, the status of a gene's mutation, and the relationship between gene expression and other variables by utilizing the database's algorithms, which include MCP-counter, quanTIseq, EPIC algorithms, and CIBERSORT. TIMER 2.0 was used to create pictorial presentation of TGFβ-1, IL19, CXCR4, BMP1, VCAN, and WNT2, to understand expression of gene profiles of all cancers better [36]. This tool allowed us to compare gene expression across various tumor types. Such insights are crucial for identifying immunologically relevant genes that may serve as potential therapeutic targets in breast cancer.

### 2.3. bc-GenExMiner

The web-based programme bc-GenExMiner v4.5 (http://bcgenex.centregauducheau.fr/BC-GEM) was created especially for the analysis of transcriptome data related to breast cancer. With the help of this platform, researchers may conduct thorough analyses of molecular characteristics and their clinical consequences, to access to a huge library of gene expression profiles obtained from several breast cancer studies [37]. During our analysis of TGFβ-1, IL19, CXCR4, BMP1, VCAN, and WNT2 expression levels, bc-GenExMiner facilitated the correlation of TGFβ-1, IL19, CXCR4, BMP1, VCAN, and WNT2 expression with several important clinical features of BC patients. In particular, we examined the relationships between TGFβ-1, IL19, CXCR4, BMP1, VCAN, and WNT2 expression and the status of the ER, PR, and HER2.

## 2.4. TISCH2

Tumor Immune Single-cell Hub 2 (TISCH2) represents a scRNA-seq database focusing on tumor microenvironment (TME) [38]. TISCH2 provides cell-type annotation at the single-cell level, facilitating the exploration of TME across various cancer types (http://tisch.comp-genomics.org/). For this study, we utilized TISCH2 to investigate the expression patterns of six key genes—TGFβ-1, IL19, CXCR4, BMP1, VCAN, and WNT2—within the TME of BC. We analysed these genes in both primary and metastatic tumor datasets to understand their differential expression and potential roles in disease progression.

## 2.5. Gene MANIA

With the use of the flexible resource GeneMANIA (https://genemania.org/), one can forecast the roles of particular genes and gene groups in a variety of species, including humans. GeneMANIA offers insights into biological pathways, functional connections, and gene interactions by utilizing a diverse range of data sources. Users can concentrate on particular kinds of interactions, like co-expression, physical interactions, common pathways, and genetic interactions, thanks to the platform's customized analysis tools(https://genemania.org/) [39]. In this study, we were able to see how TGFβ-1, IL19, CXCR4, BMP1, VCAN, and WNT2 interacted with other genes and examine several networks that demonstrated their connections to genes that were functionally relevant.

## 2.6. Enrichr

Enrichr (https://maayanlab.cloud/Enrichr/), [40,41] represents a gene set search tool that makes it possible for us to query hundreds of thousands of annotated gene sets. Enrichr distinctively furnishes information about mammalian genes and gene sets after clubbing data from several projects. The platform, available at https://maayanlab.cloud/Enrichr, furnishes several methods to compute gene set enrichment and the outcomes can be seen in several interactive waysWe analyzed Gene Ontology (GO) and KEGG pathways related to the given genes using Enrichr concentrating on three primary categories: MF, CC, and BP.

## 2.7. DISCO

DISCO represents the Deeply Integrated Human Single-Cell Omics data (https://www.immunesinglecell.org/) which has an annotated atlas containing information about various tissues like breast [42]. Various interactive plots can be generated using DISCO thus presenting an easy-to-use online tool for researchers. The users can further navigate between various parameters available within DISCO.

## 2.8. UCSC Xena

UCSC Xena represents a web-based tool for public and private, multi-omic and clinical/phenotype data (https://xena.ucsc.edu/). UCSC Xena permits a user to explore functional genomic data sets for correlations between genomic and/or phenotypic variables [43]. It provides access to publicly available datasets, including The Cancer Genome Atlas (**TCGA**), offering a robust platform for functional genomic analysis. We selected the TCGA Breast Cancer Study and analyzed the expression pattern of TGFβ-1, IL19, CXCR4, BMP1, VCAN, and WNT2 in normal, primary, and metastatic breast cancer.

## 2.9. STRING

The STRING database systematically collects and integrates protein–protein interactions, both physical interactions as well as functional links (https://string-db.org/) [44]. We analysed the association of TGFβ-1, IL19, CXCR4, BMP1, VCAN, and WNT2 using STRING. The data comes from several sources: automated text mining of the scientific literature, computational interaction predictions from co-expression, conserved genomic context, databases of interaction experiments,

and known complexes/pathways from curated sources. All of these interactions are thoroughly analyzed, scored, and followed by automatic transfer to less well-studied organisms using hierarchical orthology information. The data can be explored through the website, also programmatically and through bulk downloads as well.

### 2.10.  Density functional Theory (DFT) studies of ligand

The molecule's kinetic stability, electrical conductivity, and Density Functional theory parameters such as HOMO-LUMO are regulated by Frontier molecular orbitals (FMO). The DFT assessment of the respective compounds was carried out in the Discovery studio visualizer. The lead molecules were geometrically optimized with the Becke three-parameter hybrid functional and Lee-Yang-gradient Parr's rectified correlation functional (BLYP) without any symmetrical incommodity. The narrow HOMO-LUMO gap is associated with soft molecules which are demonstrated with great biological and greater chemical reactivity. Whereas, those molecules exhibiting higher chemical stability, have a large HOMO-LUMO gap with less biological reactivity. In this current study the Koopman's approximation was utilized to evaluate the HOMO-LUMO energy gap, HOMO, LUMO parameter [45].

### 2.11.  Molecular Docking Studies

The main aim of the study was to provide insight into the synergistic binding propensities of CXCR4 (PDBID: 4RWS) and TGFβ-1(PDB ID: 3KFD) with ligands AMD3100 and D4476 as described elsewhere [46,47]. In order to accurately fore-cast docking, the ligand binding efficiency utilizing Auto dock vs 4.2.6 was performed. After joining non-polar hydrogens, the receptor and target chemical are saved in pdbqt format so that docking studies can be conducted on the molecules in question. To develop grid boxes with precise measurements and a 0.3 Å spacing, ligands are required. Protein-ligand compound docking studies were conducted using LGA. Three duplicates of the MD studies were carried out, with each replicate having 50 solutions, 500 population members, 2500000 evaluations, a maximum generational number of 27, and all other parameters set to their default settings. Upon completion of the ligand-receptor docking, the RMSD clustering maps were produced by re-clustering with 2.0 Å clustering tolerances to identify the optimal cluster with the lowest energy score and the greatest number of populations.

## 3.  Results

### 3.1.  TGF β-1, IL19, CXCR4, BMP1, VCAN, and WNT2 are highly upregulated across pan-cancer

The TIMER 2.0 analysis revealed the overexpression of TGF β-1, IL19, CXCR4, BMP1, VCAN, and WNT2 across all PAN cancers. In comparison to non-cancerous samples, it was observed that the levels of TGFβ-1, IL19, CXCR4, BMP1, VCAN, and WNT2 were highly increased in several types of tumors. Maximum of the cancers displayed the pattern, with enhanced expression of IL19, TGFβ-1, CXCR4, BMP1, VCAN, and WNT2 protein levels in the samples of KICH, KIRC, LUAD, LUSC especially BRCA when compared to normal samples (**Fig 1**). These findings imply that the common feature of high expression of TGFβ-1, IL19, CXCR4, BMP1, VCAN, and WNT2 can be observed in many different tumors, especially in BRCA, which may indicate their universal importance as oncogenic drivers and potential therapeutic targets.

### 3.2.  Analysis of TGFβ-1, IL19, CXCR4, BMP1, VCAN, and WNT2 Expression in BC

By employing UALCAN, we examined the pattern of expression for TGFβ-1, IL19, CXCR4, BMP1, VCAN, and WNT2 in breast cancer. The research revealed that among the dysregulated genes, TGFβ-1, CXCR4, IL19, BMP1, VCAN, and WNT2 were relatively upregulated. We compared the TGFβ-1, CXCR4, IL19, BMP1, VCAN, and WNT2 expression in different subtypes of breast cancer, and the results demonstrated that BMP1, and WNT2 were consistently overexpressed in HER-2 subtype. TGFβ-1, VCAN, and IL19 are comparatively expressed more in Luminal subtype of breast cancer (**Fig 2**). Also, the expression of CXCR-4 was seen higher in TNBC subtype as compared to the other subtypes of breast

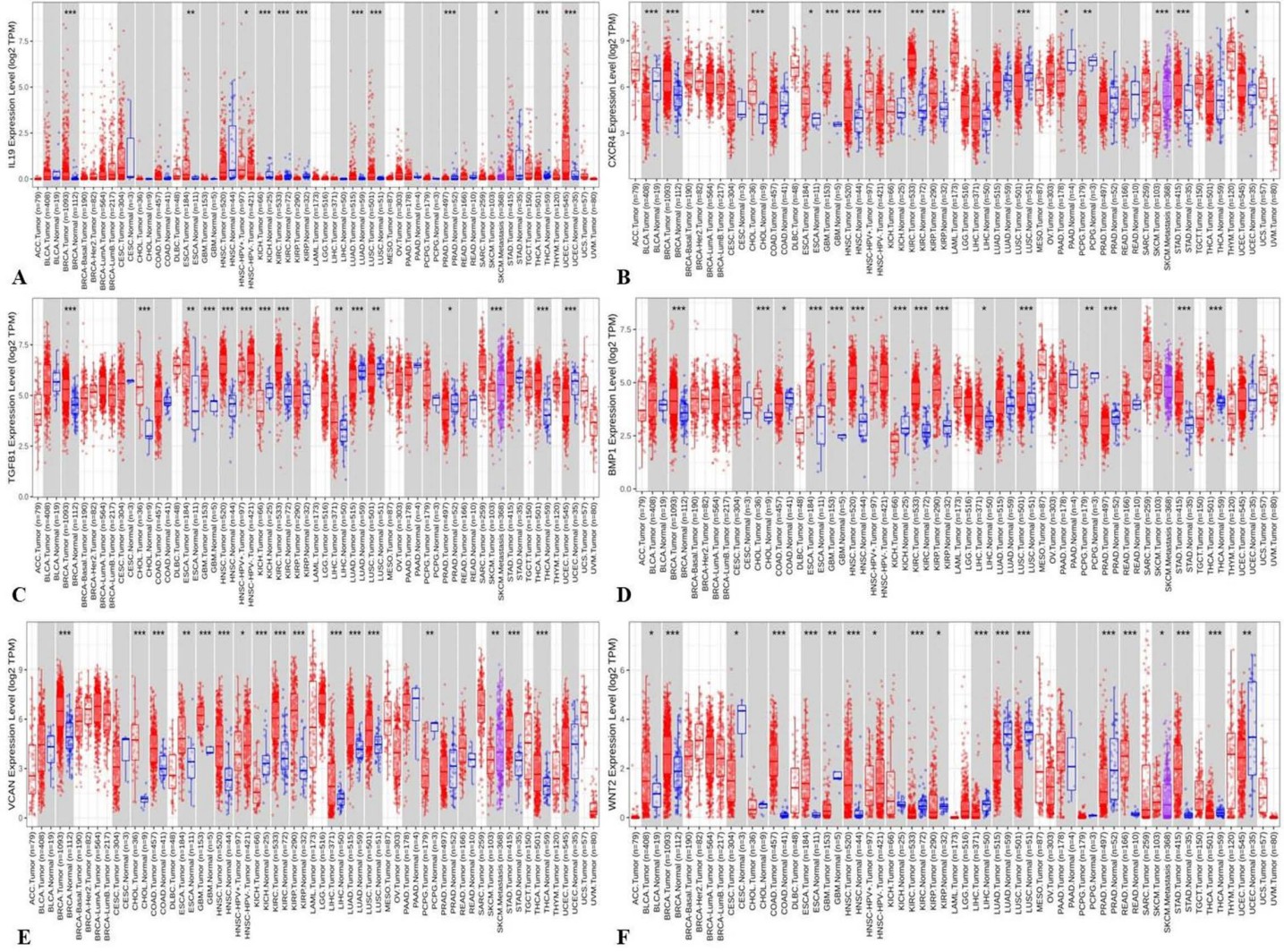

**Fig 1. Expression analysis using TIMER 2.0 revealed high expression of A) IL-19 B) CXCR4 C) TGFβ-1 D) BMP1 E) VCAN F) WNT2 genes in various cancers when compared to their respective normal tissues.** Notably, this upregulation was particularly prominent in breast invasive carcinoma (BRCA), underscoring their potential role in breast tumorigenesis.

cancer. Expression levels of these genes were also examined across various pathological stages of breast cancer, where a stage-dependent increase was observed particularly for CXCR4, BMP1, VCAN, and WNT2 (**Fig 2**).

Additionally, we used the UALCAN analysis to observe the expression profile of the genes based on nodal metastasis status. The results showed that the expression of TGFβ-1, VCAN, IL19, CXCR4, BMP1, VCAN, and WNT2 was significantly elevated in patients with nodal metastasis compared to those without, indicating their possible involvement in metastatic progression. These genes may have solid potential as biomarkers as well as therapeutic targets toward disease expansion.

### 3.3. High expression of TGFβ-1, IL19, CXCR4, BMP1, VCAN, and WNT2 in the primary breast cancer dataset

Using the TME-focused TISCH database, we evaluated the expression status of unregulated TGFβ-1, CXCR4, BMP1, VCAN, and WNT2. We looked at the TGFβ-1, CXCR4, BMP1, VCAN, and WNT2 expression profiles in a few BC datasets

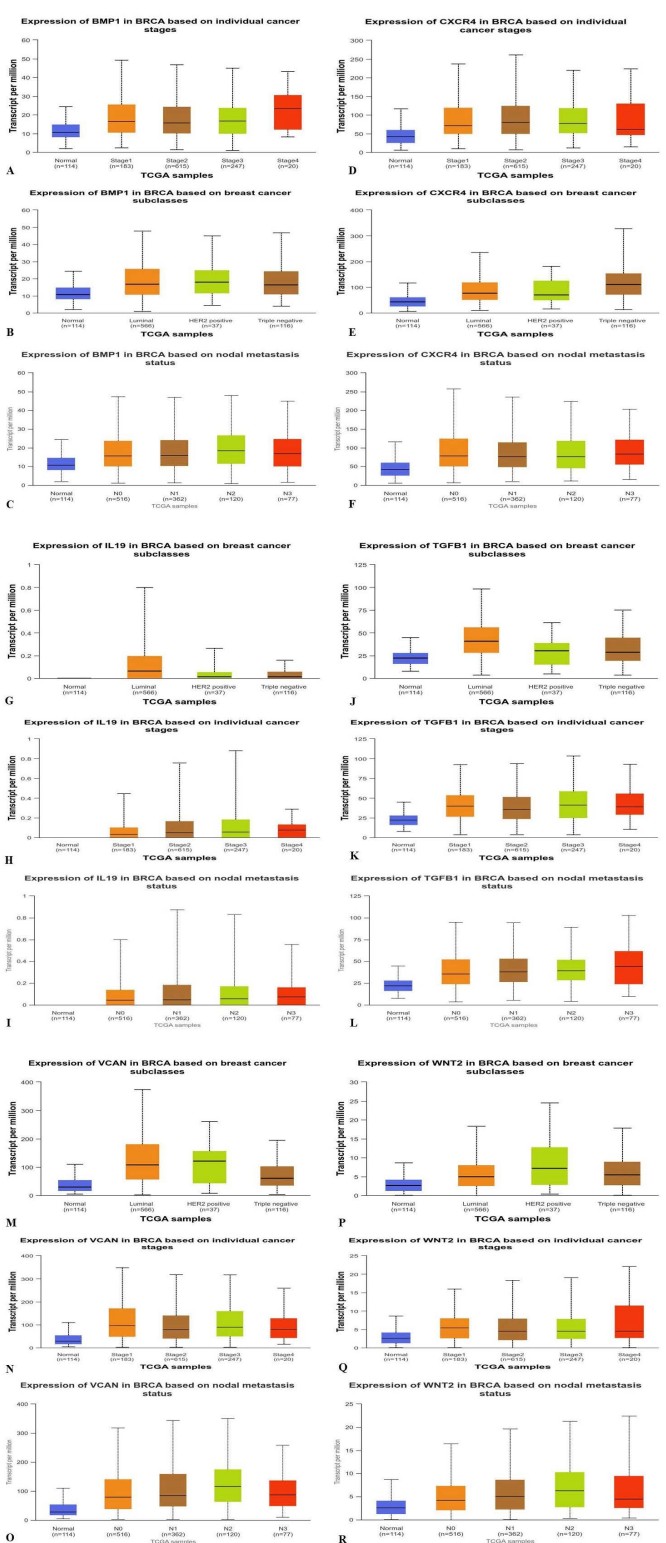

**Fig 2. Based on UALCAN analysis, the expression profile of BMP1, CXCR4, IL-19, TGFβ-1, VCAN, WNT2 genes – which were under study – was evaluated in breast cancers for three major clinical parameters: breast cancer subtype, individual cancer stages, and nodal metastasis status.** UALCAN Analysis of BMP 1 (A, B, C); CXCR4 (D, E, F); IL19 (G, H, I); TGFβ-1(J, K, L); VCAN (M, N, O); and WNT2 (P, Q, R) revealed higher expression in BRCA based on breast cancer subclass, individual cancer stages and nodal metastasis status.

related to primary and metastatic tumors. TGFβ-1, CXCR4, BMP1, VCAN, and WNT2 are more expressed in original tumors compared to metastatic tumors, according to the data. Additionally, there was variation in the expression profile of the main tumor cell population. CXCR4 was one of the genes that had a notable rise in expression among the others, particularly in the primary dataset (Fig 3).

### 3.4. Gene-gene interaction of investigated genes

By exploring the GeneMANIA portal, we examined the association between the investigated genes in the gene network. The investigated genes displayed significant interactions with ADAM12 followed by PAMR1, MMP11, VSTM4, and quite a few other genes as shown in Fig 4. The network interaction of genes was based on physical connections, co-expression, co-localization, and genetic interaction pathways. The selected genes are functionally related to each other in some of the most important oncogenic systems, possibly cooperating in pathways relevant to tumor progression, extracellular matrix remodeling, and metastasis, and so emerging as co-targetable nodes in breast cancer.

### 3.5. Expression of investigated genes in varied types of breast cells

We explored the DISCO database to find out the expression levels of investigated genes in various cells of the breast. The results revealed that TGFβ-1, IL19, CXCR4, BMP1, VCAN, and WNT2 expression is enriched in several different breast tissues/cells (Fig 5). TGFβ-1 was highly enriched in CD4 T cells, CD8 T cells, CXCL 14 mammary basal cells, KRT6B mammary basal cell, venous EC, arterial EC, capillary EC, and lymphatic EC (Fig 5B). The expression of CXCR4 was highly enriched in macrophages, monocytes, B cells, dendritic cells, NK cells, T reg cells, CD8, CD4, fibroblasts etc (Fig 5C). The expression of BMP1 was seen higher mostly in pericytes and fibroblasts (Fig 5D). The expression of VCAN was seen higher mostly in fibroblasts, venous EC, and monocytes (Fig 5E). WNT2 was highly enriched in fibroblast cells (Fig 5F).

### 3.6. Gene ontology analysis

By employing Enrichr, an online tool for GO investigation, it was revealed that the various processes in which TGFβ-1, IL19, CXCR4, BMP1, VCAN, and WNT2 were seen to be actively engaged among the MF, including Cytokine Activity, Receptor Ligand Activity, Metallopeptidase Activity, C-C Chemokine Binding, Type II Transforming Growth Factor Beta Receptor Binding, Type I Transforming Growth Factor Beta Receptor Binding, Chemokine Binding, Transforming Growth Factor Beta Receptor Binding, Chemokine Receptor Activity (Fig 6A). TGFβ-1, IL19, CXCR4, BMP1, VCAN, and WNT2 were implicated in several biological processes, including Positive Regulation of Glial Cell Differentiation, Regulation of Chemotaxis, and Response to Transforming Growth Factor Beta (Fig 6B). Further analysis revealed that TGFβ-1, IL19, CXCR4, BMP1, VCAN, and WNT2 were abundant in Collagen-Containing Extracellular Matrix, Golgi Lumen, Perineuronal Net, and Perisynaptic Extracellular Matrix in the GO analysis' CC terms (Fig 6C). Also, the KEGG analysis revealed that the investigated genes were highly enriched in pathways like cytokine-cytokine interaction, hepatocarcinoma, pathways in cancer, gastric cancer, etc. (Fig 6D–E).

### 3.7. Comparison of expression analysis of investigated genes

UCSC Xena was investigated to find the expression of the investigated genes and to compare their expression with each other. The heat map showed the expression analysis of TGFβ-1, IL19, CXCR4, BMP1, VCAN, and WNT2. Among all the six genes, TGFβ-1 showed high expression in primary tumors followed by IL-19, BMP1, and CXCR4 (Fig 7). This observation further emphasizes the heightened expression of TGFβ-1 and related genes in primary breast tumors acting as a plausible mechanism for tumor development alongside therapeutic diagnosis.

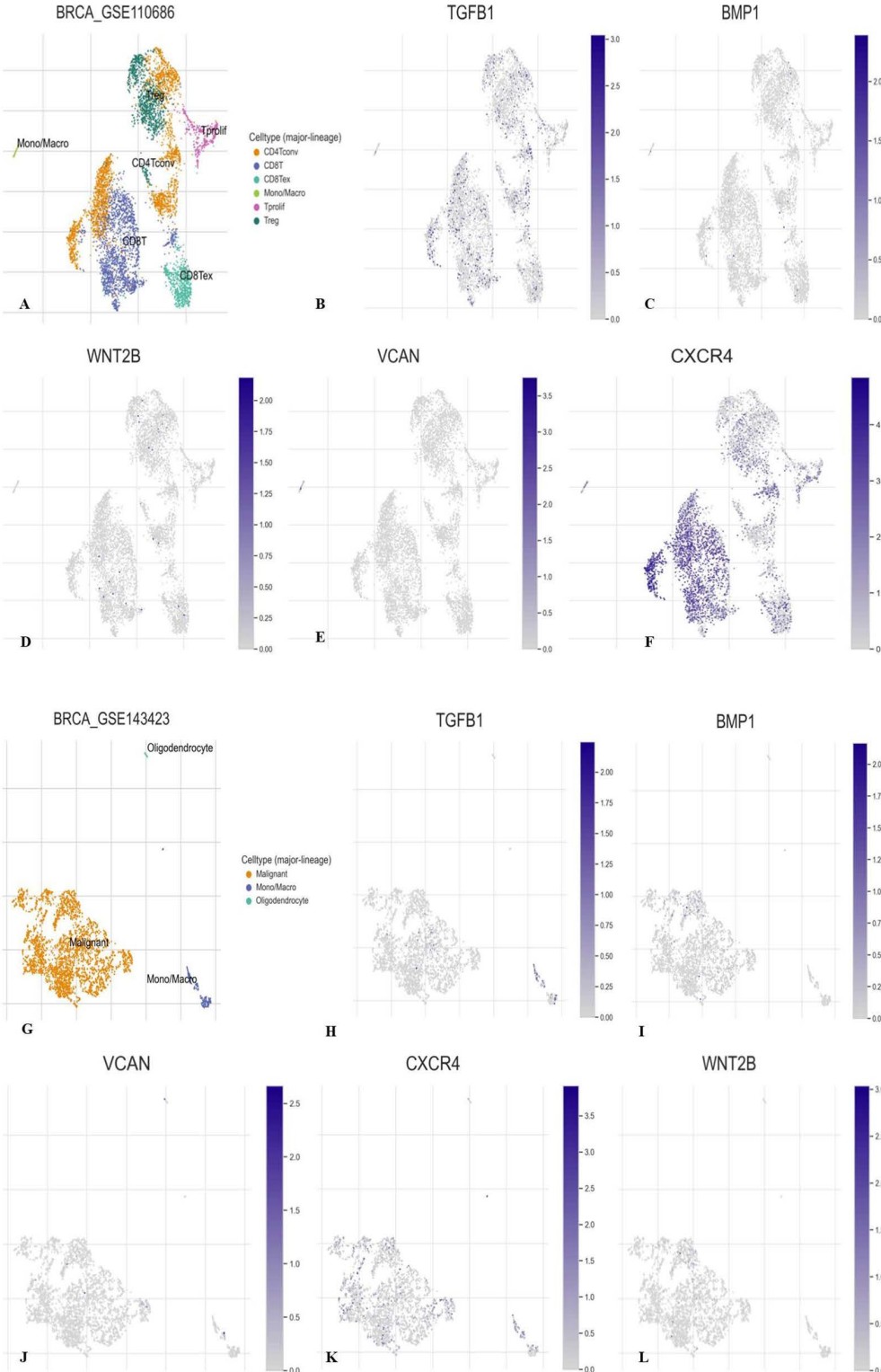

**Fig 3. Cell-type specific expression patterns of TGFβ-1, BMP1, WNT2B, VCAN, and CXCR4 in primary and metastatic tumor datasets (TISCH2 analysis).** (A-F) Analysis of TGFβ-1, BMP1, WNT2B, VCAN, and CXCR4, in primary datasets. (G-L) Analysis of TGFβ-1, BMP1, VCAN, and CXCR4, and WNT2B in metastatic datasets. TISCH2 Analysis revealed the relative expression of CXCR4, BMP1, TGFβ-1, VCAN, and WNT2B in various cell

types (major lineages) considering primary and metastatic datasets. The highest expression was seen in CXCR4 in the primary dataset. TISCH2 analysis revealed delineating cell type-oriented expression patterns across the major immune and stromal lineages: epithelial cells, fibroblasts, T cells, and macrophages. Among them, CXCR4 clearly rose highest with expression levels in the primary tumor datasets, hinting at an important role in early-stage tumor-immune interactions and in shaping the tumor microenvironment.

### 3.8. Investigating the impact of explored genes on Breast cancer dysregulated pathways

We explored bc-GenExMiner for analyzing the impact of TGFβ-1, IL19, CXCR4, BMP1, VCAN, and WNT2 on the dysregulated pathways in different subtypes of BC. The results revealed that TGFβ-1, IL19, CXCR4, BMP1, VCAN, and WNT2 showed correlation with WNT2B, AKT3, PIK3CA, and NOTCH4 (**Fig 8**). The expression pattern of TGFβ-1, IL19, CXCR4, BMP1, VCAN, and WNT2 in different molecular subclasses of Breast Cancer revealed that TGFβ-1, IL19, CXCR4, BMP1, VCAN, and WNT2 mRNA levels are higher in luminal A followed by luminal B, HER2 and Basal-like respectively.

### 3.9. Protein-protein Interaction Analysis of Investigated Genes

We created a network of protein-protein interactions by putting together the co-expressed genes with multiple protein-protein connections in a STRING database. Further analysis of the PPI network (**Fig 9A**) showed that the average node degree was 11, the predicted number of edges was 20, the average local clustering coefficient was 0.745, and the PPI

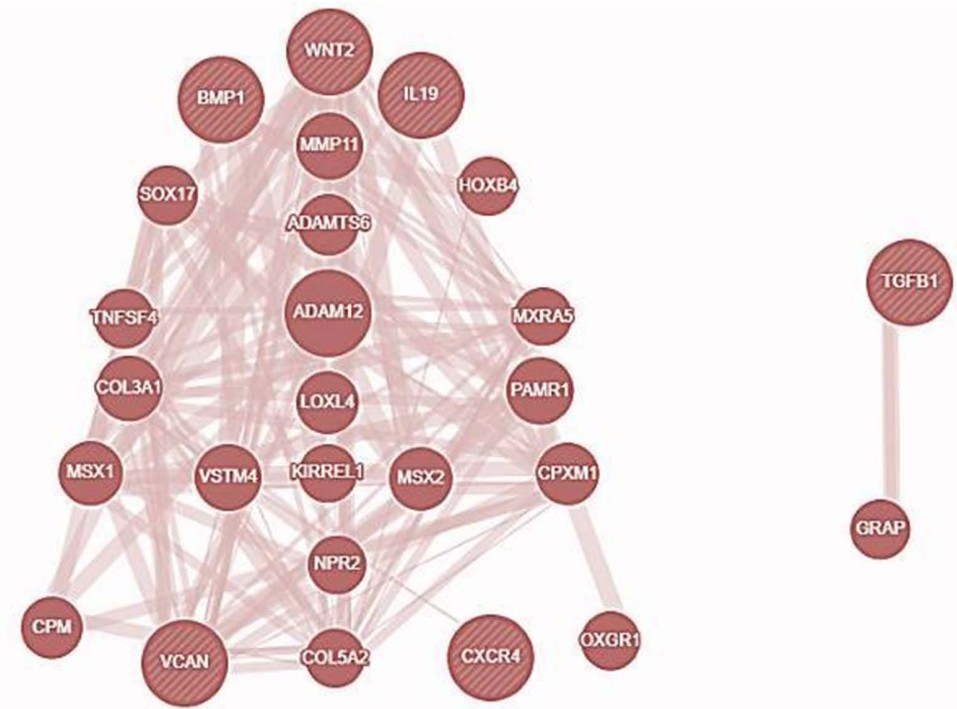

**Fig 4. Gene Mania revealed the association of the selected six genes (TGFβ-1, IL19, CXCR4, BMP1, VCAN, and WNT2) with one another, and with other significant genes that play important roles in cancer including ADAM12, PAMR1, MMP11, and VSTM4.** Co-localization, co-expression patterns, common pathways, and physical protein-protein interactions served as the foundation for these connections.

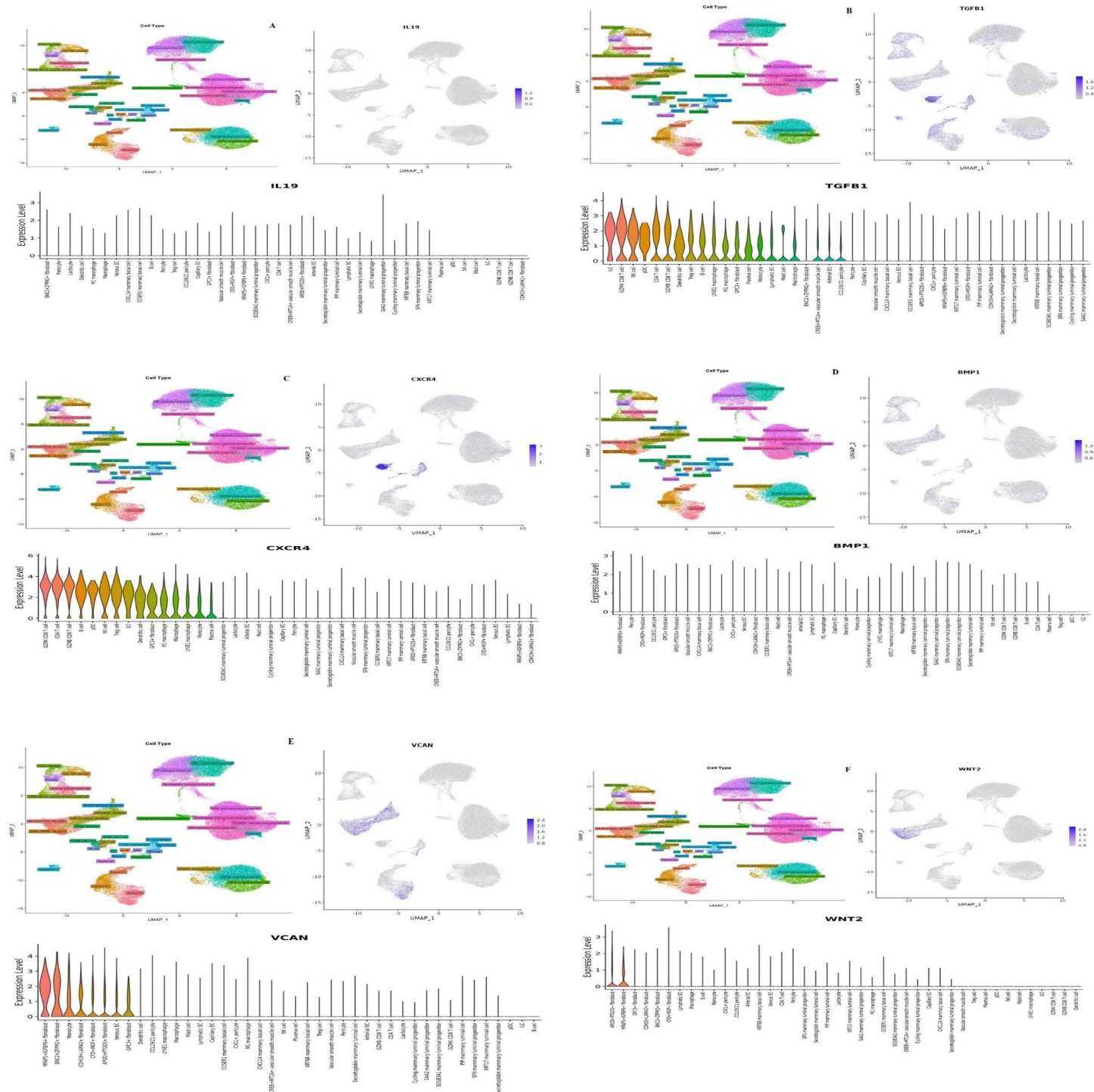

**Fig 5. Analysing the expression level of (A) IL19, (B)TGFβ-1, (C) CXCR4, (D) BMP1, (E) VCAN, and (F) WNT2 through the DISCO database.**

enrichment p-value was 4.53e-05. Cytohubba was used to identify the top 10 hub genes in the network based on degree score, as Fig 9B illustrates. The top 10 genes in the network were BGN, SDC4, *POSTN, TGFβ-1, IL19, CXCR4, BMP1, VCAN, WNT2,* and *GDF5.*

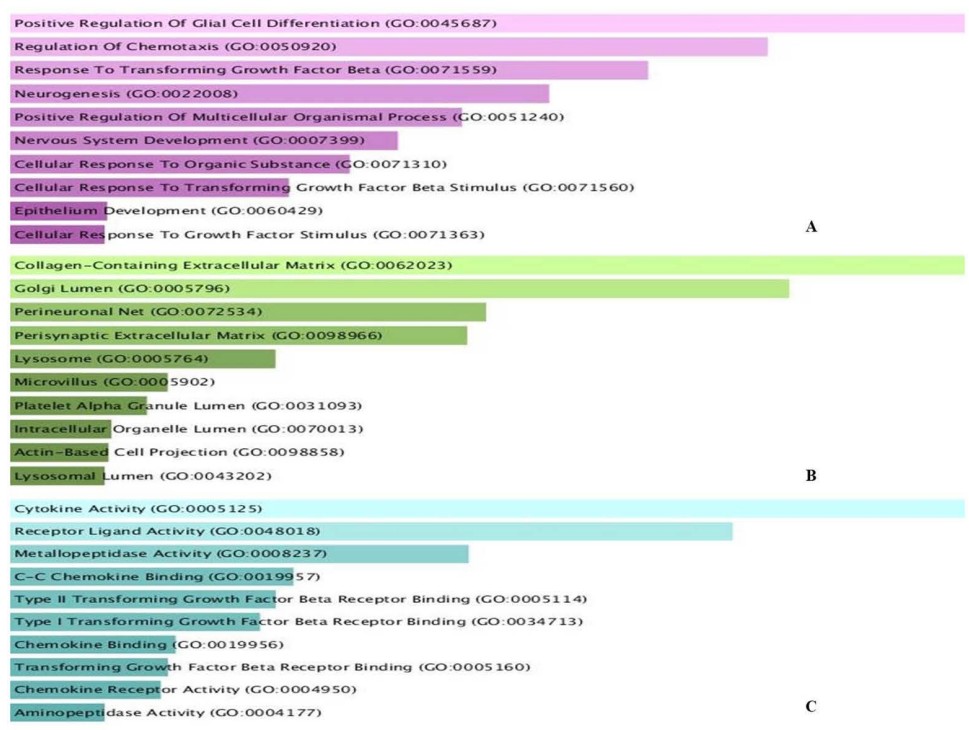

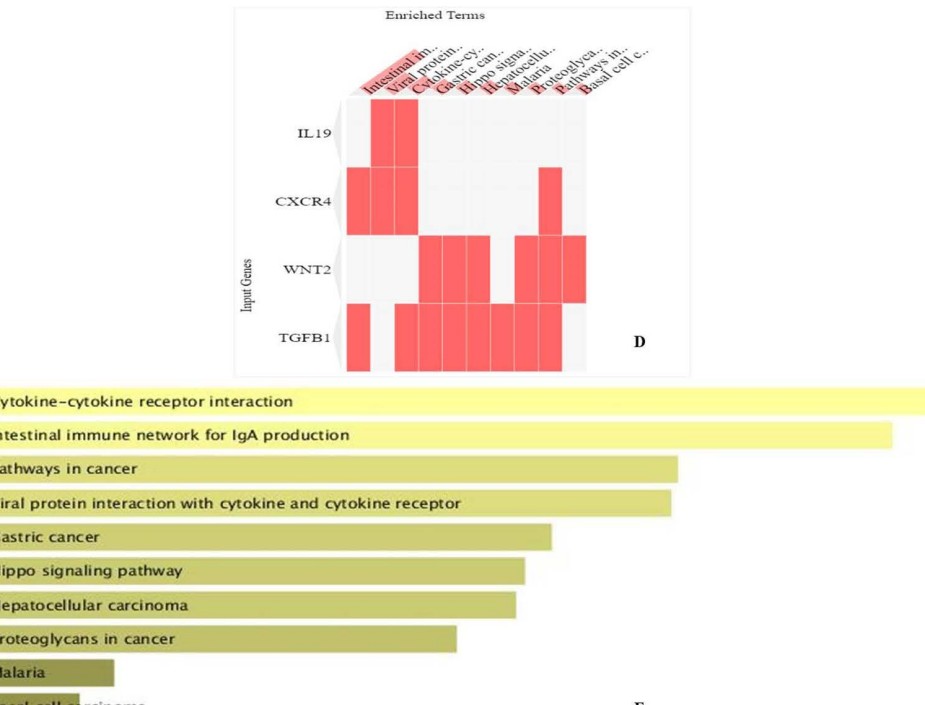

**Fig 6. Analysis of the selected genes through the Enrichr database.** (A) Molecular function, (B) Biological function, (C) Cellular compartments, (D-E) KEGG analysis of TGFβ-1, IL19, CXCR4, BMP1, VCAN, and WNT2.

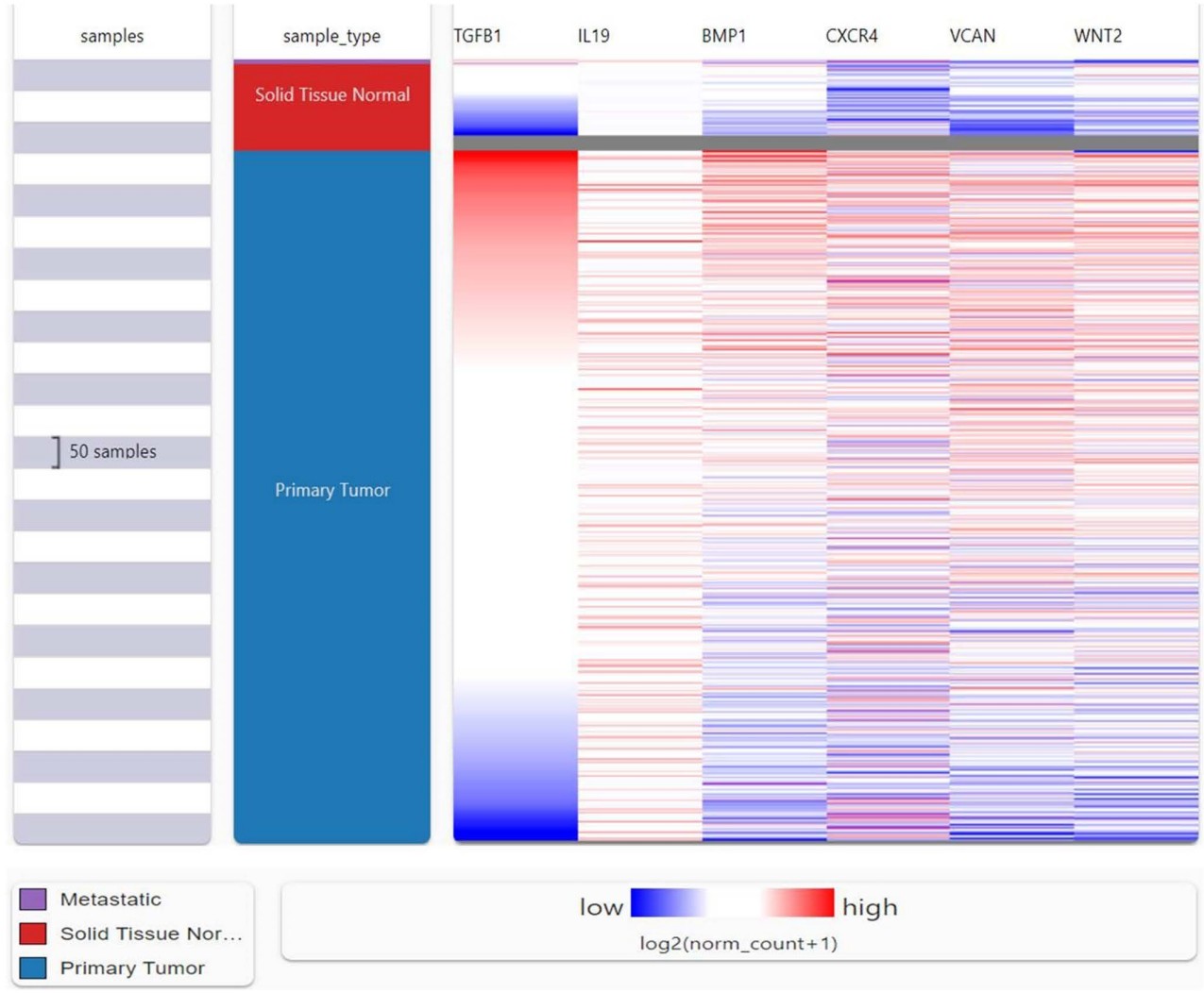

**Fig 7. Comparison of selected genes and the relative expression between primary tumor and normal solid tissue using the UCSC Xena database.** Consistent and significant upregulation of all six genes was observed within the primary tumor samples when compared to normal tissues. Such evidence goes to underscore their oncogenic potential and possible involvement in tumor initiation, progression, and microenvironmental remodelling.

### 3.10. Density functional theory

Among the six investigated genes, our study revealed significant results for *TGFβ-1* and *CXCR4*. Therefore, targeting *TGFβ-1* and *CXCR4* with the potential drug inhibitors may prove a significant approach to target breast cancer. In this regard, we selected two drug molecules namely D4476 and AMD3100 to target the respective significant genes. In the gas phase, the DFT parameter namely HOMO & LUMO were calculated to evaluate the chemical reactivity of a molecule along with the binding energy which is designated as a disparity between the initial reactive structure and final optimized structure which showed large deviations. In this study, the D4476 molecule demonstrated a total energy of − 1331.5 Ha, binding energy of −8.77 Ha, HOMO energy of −0.1644 Ha, LUMO energy of −0.080 Ha, and the band gap energy was calculated as 0.08427 Ha which was equal to 2.2931 eV (**Fig 10A**). Whereas, another ligand AMD3100 showed a total energy of about −1536.55 Ha, binding energy of −13.65 Ha, HOMO energy of −0.1356 Ha, LUMO energy of −0.01293 Ha,

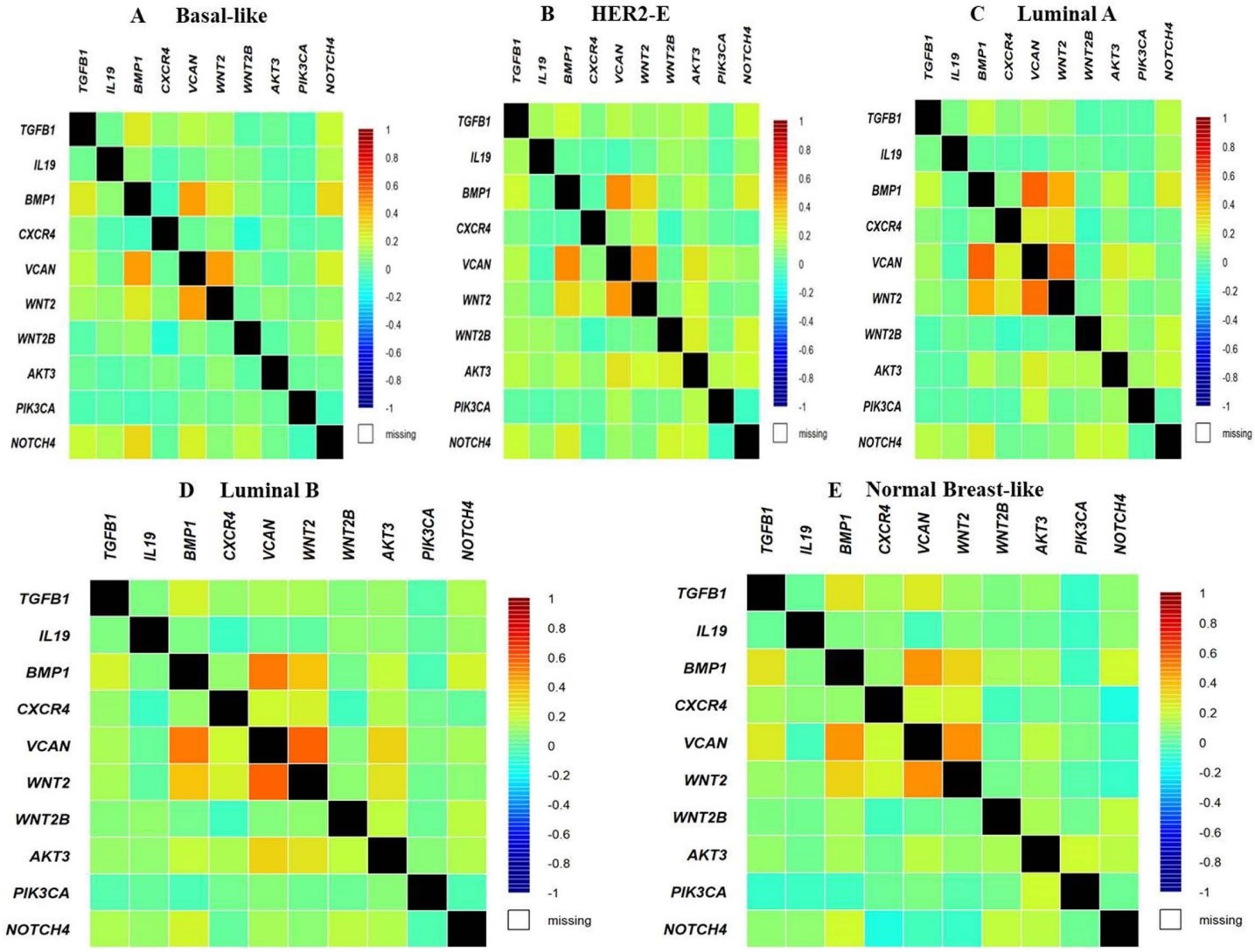

**Fig 8. Correlation of TGFβ-1, IL19, CXCR4, BMP1, VCAN, and WNT2 with dysregulated pathways like WNT 2B, AKT3, PIK3CA, NOTCH4 in different sub classes of Breast Cancer.**

and the band gap energy was calculated as 0.122735 Ha which was equal to 3.339 eV. Here, Ha stands for Hatree energy and 1 Ha is equal to 27.2114 eV (**Fig 10B**). Moreover, the highest occupied molecular orbital and lowest unoccupied molecular orbital are certain crucial parameters which describes a molecule energy differences and the ability to accept and donate electrons. The other essential parameters for effective quantum chemical analysis of electronic energies are described in **Table 1** namely – hardness (η), softness (S), electronegativity (χ), and electrophilicity (ω). The biological activity of a molecule was estimated through the Molecular electrostatic potential (MEP) tool which demonstrates the electrostatic potential surfaces for the gaseous phases of the molecule. The MEP data is represented in a universal reverse rainbow color bar form with the envelope's iso-value set to 0.02 a.u. The highest negative potential is designated with red color, whereas the positive potential areas are designated with blue color, and zero potential regions are illustrated with green color. The molecular electrostatic potential analysis of the DFT optimized structure of the molecule- D4476, AMD3100 is exhibited in **Fig 10C–D**.

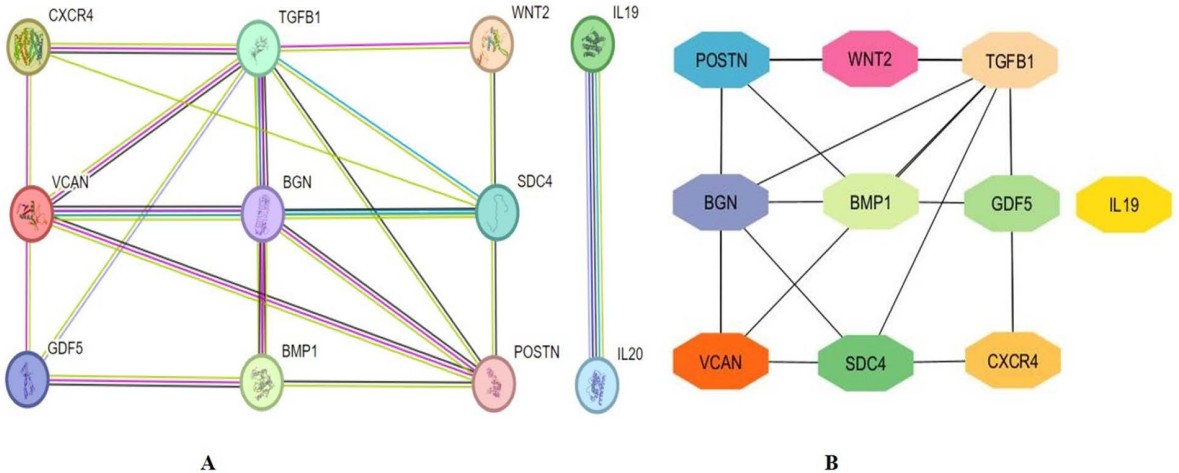

**Fig 9. PPI network construction and hub gene analysis.** A) The PPI analysis of TGFβ-1, IL19, CXCR4, BMP1, VCAN, and WNT2, B) The identification of the top 10 hub genes within the network was accomplished through the utilisation of Cytohubba tool. TGFβ-1, IL19, CXCR4, BMP1, VCAN, WNT2, POSTN, BGN, GDF5, and SDC4 are the genes included in the grid.

### 3.11. Molecular docking analysis

All the binding energy scores are computed starting with the 95% best cluster, which is within the lowest RMSD of 0.25 Å. AMD3100 demonstrated a strong binding affinity with CXCR4 with the lowest binding energy (ΔG – 5.1 kcal/mol) and inhibitory concentration, Ki (11.4 µM) (**Fig 11A**). At the protein's binding cavity, Leu20 participates in a pi-alkyl interaction with the ligand AMD3100. On the other hand, D4476 displayed the lowest inhibitory concentration, Ki (9.1 µM), and energy ΔG – 5.4 kcal/mol. During the contact of the ligand, D4476 made hydrogen bonding with Lys26, Tyr21, and Asp27 residues (**Fig 11B**). Interestingly, multiple ligands simultaneous docking (MLSD) considering both the ligands docked at the same time at the same position exhibited binding energy lower than −6.4 kcal/mol. This might have happened due to the competitive behavior of AMD3100 and D4476 for the binding cavity to interact. Both the ligand made a polar contact forming hydrogen bond with Leu20 (**Fig 11C**). MLSD interpreted that a drug consortium can be utilized to achieve a compounding effect in the ailment of the disease. For TGFβ-1, AMD3100 exhibited binding energy −5.4 kcal/mol with an inhibitory concentration of 9.1 µM. Phe1114 and Arg1137 have forming pi-pi stacked and pi-alkyl interactions respectively with the binding cavity residues (**Fig 11D**). Ligand D4476 bound more tight interaction with TGFβ-1 exhibiting binding energy −6.1 kcal/mol and interacting with Met1106 as pi-sulfur and Phe1114 pi-pi stacked interactions with the binding cavity residues (**Fig 11E**). While MLSD of D4476 and AMD3100 with TGFβ-1 has shown comparable binding energy of −5.8 kcal/mol having polar contact with Ser1136 (**Fig 11F**). Therefore, from this docking study, it can be signified that both AMD3100 and D4476 have a higher propensity for CXCR4 and TGFβ-1 receptors and their consortium could invoke a significant level of inhibition to the target proteins.

### 3.12. Molecular dynamics simulation (MDS)

To reveal the stability and convergence of TGFβ-1 and CXCR4 with drugs AMD3100 and D4476, MDS studies were carried out. Simulation of 200 ns revealed stable conformation while linking the RMSD values. The RMSD of the Cα-backbone of TGFβ-1 bound to AMD3100 unveiled a deviation of 2.8 Å, while the D4476 bound protein exhibited 2.8 Å (**Fig 12A**). All the RMSD values are within a suitable range which should be below 3 Å. A stable RMSD plot during simulation indicates good convergence and stable conformations. Thus, it can be proposed that drugs bound to TGFβ-1are quite

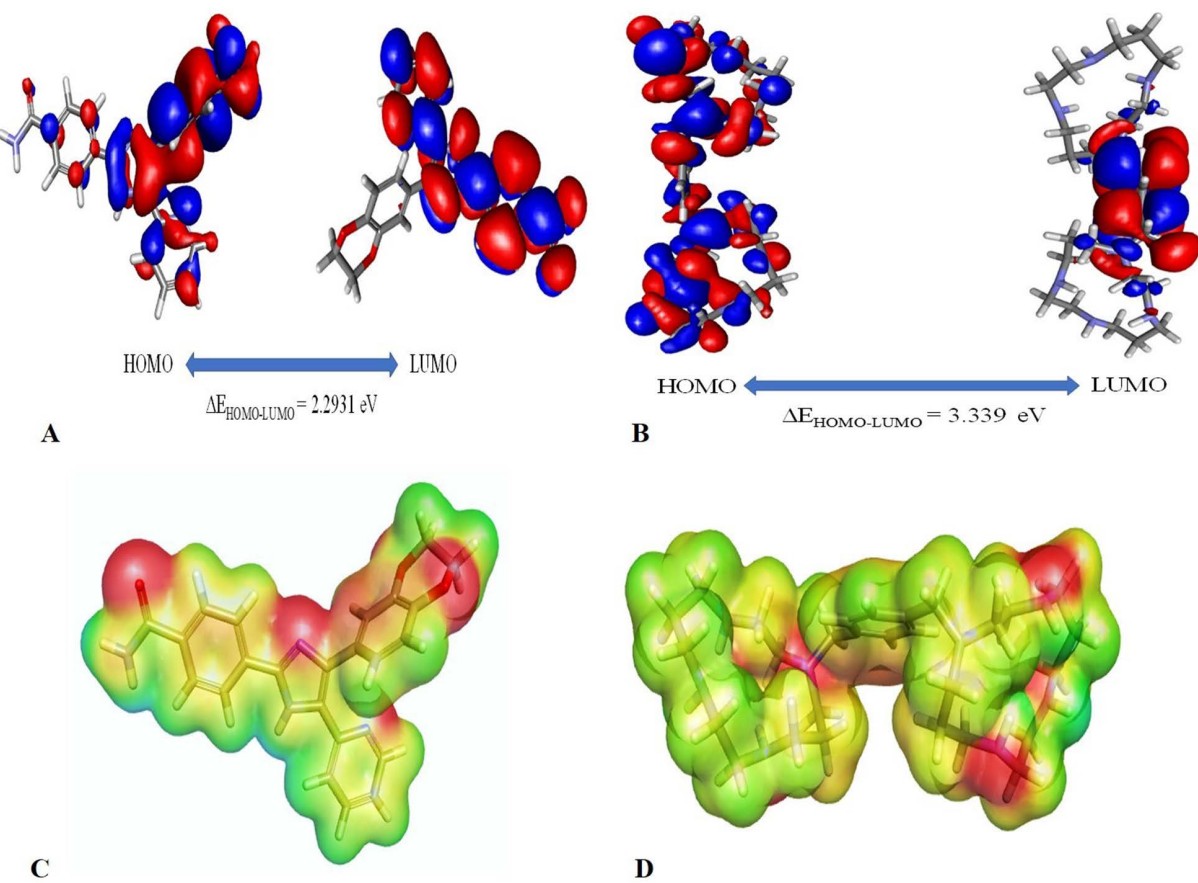

**Fig 10. HOMO and LUMO surfaces of (A) D4476 and (B) AMD3100 exhibiting high electronegative surface (red) and lower negative surface (blue).** Molecular electrostatic potential (C) D4476 and (D) AMD3100.

**Table 1. Essential parameters of D4476 and AMD3100.**

| Name of ligands | Conceptual DFT parameters (Ha) | | | |
|---|---|---|---|---|
| | Chemical hardness (η) | Chemical softness (S) | Electronegativity (χ) | Electrophilicity (ω) |
| D4476 molecule | 0.0422 | 23.696 | 0.122 | 0.1753 |
| AMD3100 molecule | 0.0613 | 16.313 | 0.074 | 0.0440 |

stable in complex due to the higher affinity of the ligand. Similarly, CXCR4 bound to AMD3100 exhibited an RMSD value of 2.91 Å, and with D4476 bound protein showed an RMSD of 3.0 Å (**Fig 12B**). The plot for root mean square fluctuations (RMSF) displayed small spikes of fluctuation in TGFβ-1 protein with AMD3100 where no significant spikes are observed except at 150−155 residues which may be due to higher flexibility of the residues (**Fig 12C**). TGFβ-1 with D4476 exhibits fluctuation at 30−45 and 150−175 residue positions (**Fig 12C**). Protein CXCR4 bound to AMD3100 exhibited no significant fluctuations indicating rigid protein conformation while bound onto ligand (**Fig 12D**). Whereas, D4476 bound protein is displaying residual fluctuations at 180−230 residues (**Fig 12D**). Fig 12C and 12D show that most residues fluctuated less during the 200 ns simulation, suggesting that the amino acid conformations remained stable. Consequently, it may be

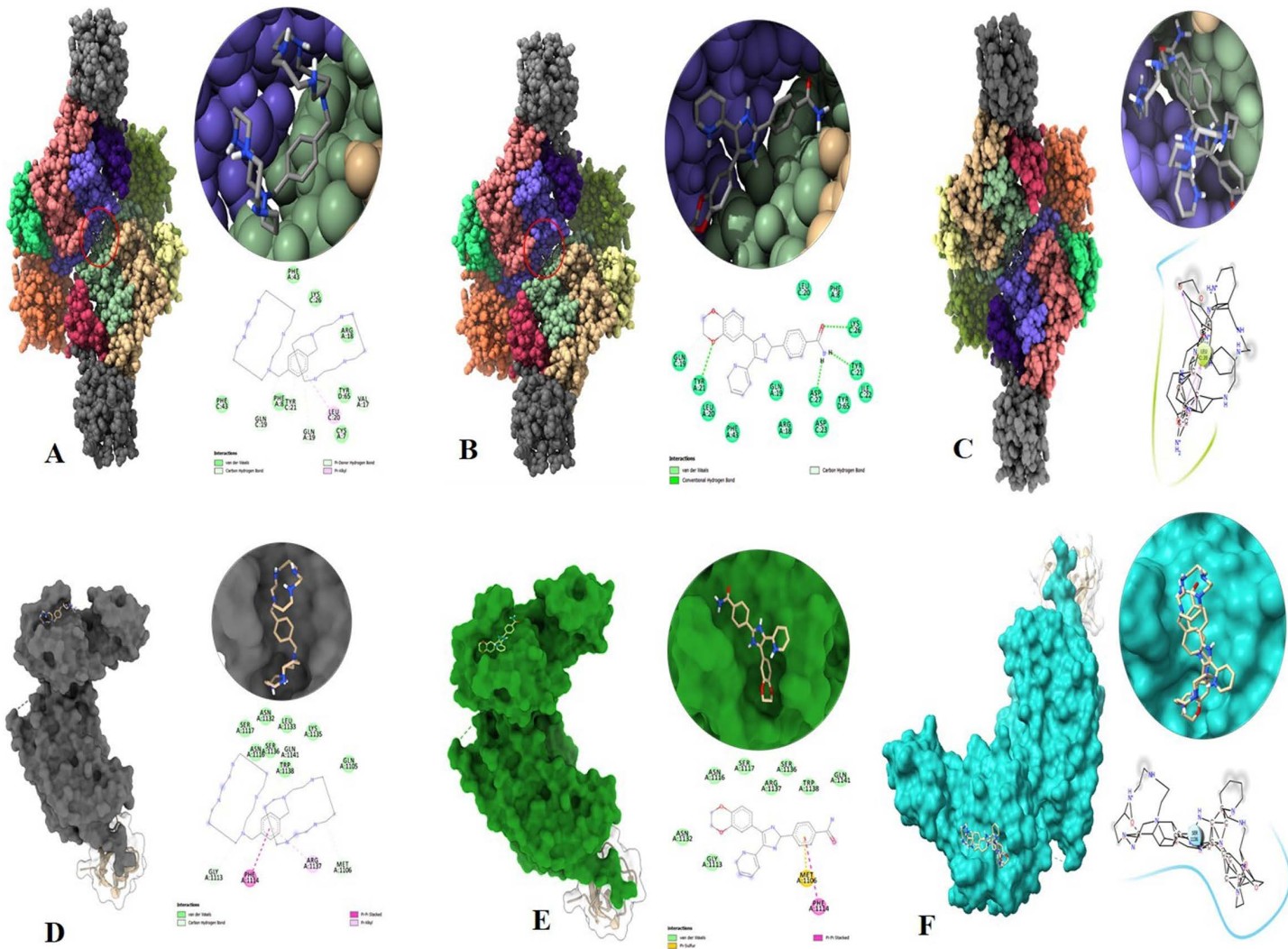

**Fig 11. Molecular docking studies and poses (A) CXCR4 interaction with AMD3100, Left Panel: Surface view of the CXCR4 protein with its deep core accommodating the ligand AMD3100.** Upper Right Panel: Zoomed view of the binding cavity in CXCR4 showing AMD3100 in place. Lower Right Panel: 2D interaction diagram highlighting how AMD3100 interacts with specific amino acid residues within the binding cavity of CXCR4. (B) CXCR4 interaction with D4476, Left Panel: Surface view of CXCR4 with the deep core accommodating the ligand D4476. Upper Right Panel: Close-up view of the binding cavity in CXCR4 with D4476 positioned within. Lower Right Panel: 2D interaction diagram showing the interactions between D4476 and the amino acids in the binding cavity of CXCR4. (C) MLSD Pose of CXCR4 with a synergy of AMD3100+D4476, Left Panel: Surface view of CXCR4 hosting both ligands, AMD3100 and D4476, in a multiple ligand simultaneous docking (MLSD) configuration. Upper Right Panel: Detailed view of the binding cavity in CXCR4 accommodating both AMD3100 and D4476. Lower Right Panel: 2D interaction diagram showing the complex network of interactions between CXCR4 and the two ligands. (D) TGFβ-1 interaction with AMD3100, Left Panel: Surface view of the TGFβ-1 protein with its deep core accommodating the ligand AMD3100. Upper Right Panel: Zoomed view of the binding cavity in TGFβ-1 showing AMD3100 in place. Lower Right Panel: 2D interaction diagram highlighting how AMD3100 interacts with specific amino acid residues within the binding cavity of TGFβ-1 (E) TGFβ-1 interaction with D4476, Left Panel: Surface view of TGFβ-1 with the deep core accommodating the ligand D4476. Upper Right Panel: Close-up view of the binding cavity in TGFβ-1 with D4476 positioned within. Lower Right Panel: 2D interaction diagram showing the interactions between D4476 and the amino acids in the binding cavity of TGFβ-1 (F) MLSD Pose of TGFβ-1 with synergy of AMD3100+D4476, Left Panel: Surface view of TGFβ-1 hosting both ligands, AMD3100 and D4476, in a multiple ligand simultaneous docking (MLSD) configuration. Upper Right Panel: Detailed view of the binding cavity in TGFβ-1 accommodating both AMD3100 and D4476. Lower Right Panel: 2D interaction diagram showing the complex network of interactions between TGFβ-1 and the two ligands.

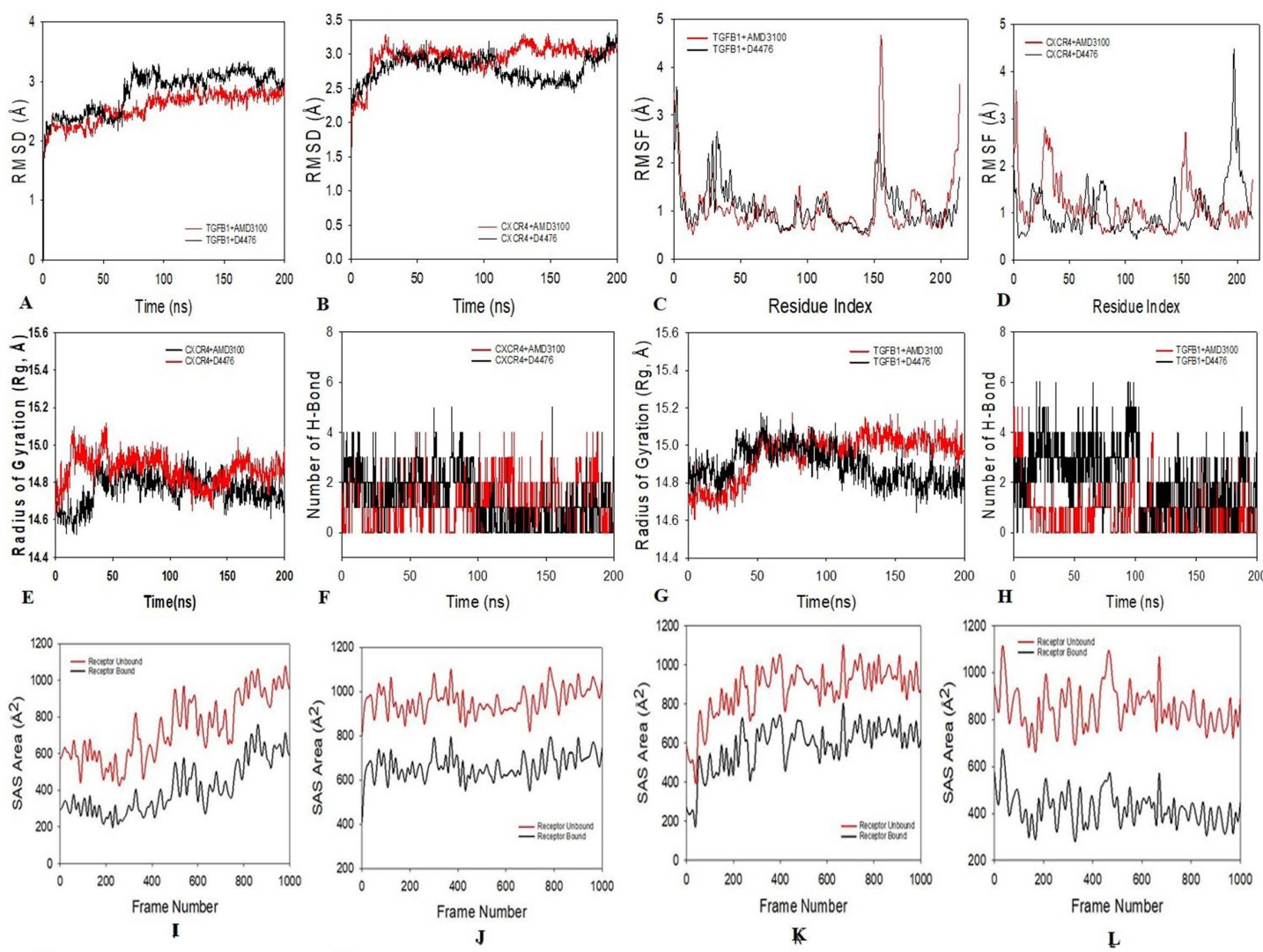

**Fig 12. MDS examination of 200 ns trajectories of (A) Cα backbone RMSD of TGFβ-1+AMD3100 (red), TGFβ-1+D4476 (black), (B) Cα backbone RMSD of CXCR4+AMD3100 (red) and CXCR4+D4476 (black).** (C) RMSF of Cα backbone of TGFβ-1+AMD3100 (red), TGFβ-1+D4476 (black), (D) Cα backbone RMSF of CXCR4+AMD3100 (red) and CXCR4+D4476 (black). (E) The radius of gyration (Rg) of Cα backbone of CXCR4+AMD3100 (Black) and CXCR4+D4476 (Red), (F) Formation of hydrogen bonds in CXCR4+AMD3100 (red) and CXCR4+D4476 (black). (G) Cα backbone radius of gyration (Rg) of TGFβ-1+AMD3100 (red), TGFβ-1+D4476 (black), (H) Development of hydrogen bonds in TGFβ-1+AMD3100 (red), TGFβ-1+D4476 (black). The solvent-accessible surface area of (I) TGFβ-1+AMD3100, (J) TGFβ-1+D4476, (K) CXCR4+AMD3100, and (L) CXCR4+D4476.

inferred from RMSF plots that, in ligand-bound conformations, the protein structure is stiff during simulation. The protein's compactness is measured by its radius of gyration (Rg). Here in this study, Rg values for CXCR4 bound AMD3100 exhibited to be stable from 14.8 to 14.9 Å and that to for bound D4476 exhibited 14.7 to 14.76 Å (**Fig 12E**). TGFβ-1 Cα-backbone bound to AMD3100 displayed a constant radius of gyration (Rg) from 14.8 to 15.0 Å (**Fig 12G**). On the other hand, a pattern was observed for D4476 bound protein from 14.8 to 14.7 Å (**Fig 12G**). Significantly stable gyration (Rg) indicates a highly compact orientation of the protein in a ligand-bound state. The number of hydrogen bonds between protein and ligand is indicative of the complex's substantial contact and stability. Hydrogen bond count between TGFβ-1

**Table 2. Binding free energy components for the TGFβ-1+AMD3100, TGFβ-1+D4476, CXCR4+AMD3100, and CXCR4+D4476 computed by MM-GBSA.**

| Energies (kcal/mol) | TGFβ-1+D4476 | CXCR4+D4476 | TGFβ-1+AMD3100 | CXCR4+AMD3100 |
|---|---|---|---|---|
| $\Delta G_{bind}$ | −78.86 | −113.00 | −76.82 | −98.07 |
| $\Delta G_{bind}$Lipo | −21.91 | −28.60 | −22.43 | −25.25 |
| $\Delta G_{bind}$vdW | −44.45 | −103.35 | −42.90 | −83.97 |
| $\Delta G_{bind}$Coulomb | 36.65 | −24.93 | −63.27 | −20.91 |
| $\Delta G_{bind}$H$_{bond}$ | −4.01 | −9.11 | −4.08 | −8.67 |
| $\Delta G_{bind}$SolvGB | 43.81 | 41.59 | 69.16 | 33.81 |
| $\Delta G_{bind}$Covalent | 3.19 | 14.41 | 2.56 | 11.68 |
| $\Delta G_{bind}$Packing | −5.42 | −4.0 | −5.06 | −4.76 |

and AMD3100 displayed significant (2) numbers and with D4476 (3) numbers are observed (**Fig 12H**), between AMD3100 and CXCR4 (2 numbers) and with D4476 (2 numbers) are observed, throughout the simulation time 200 ns (**Fig 12F**). Followed by Rg analysis, a similar pattern was also witnessed in Solvent accessible surface area (SASA) in both ligand-bound and unbound states. This is evident from Fig 12(I-L), that in the unbound state of AMD3100 and D4476 to receptor the protein TGFβ-1 and CXCR4 showed high surface area available to solvent in all the cases.

### 3.13. Molecular mechanics generalized born surface area (MM-GBSA) calculations

By employing the MD simulation trajectory, the binding free energy along with other contributing energy in the form of MM-GBSA were known for each TGFβ-1+AMD3100, TGFβ-1+D4476, CXCR4+AMD3100, and CXCR4+D4476. The outcomes (**Table 2**) indicated that the highest contribution to $\Delta G_{bind}$ in the stability of the simulated complexes was due to $\Delta G_{bind}$Coulomb, $\Delta G_{bind}$vdW, and $\Delta G_{bind}$Lipo, while, $\Delta G_{bind}$Covalent and $\Delta G_{bind}$SolvGB contributed to the instability of the resulting complexes. The TGFβ-1+AMD3100 and CXCR4+AMD3100 complexes revealed enhanced binding free energies relatively. These outcomes supported the potential of AMD3100 molecule with TGFβ-1 and CXCR4, revealed the competence in binding to the chosen protein, and the capacity to develop stable protein-ligand complexes.

### 3.14. Pearson correlation matrix analysis

The Pearson correlation matrix actually portraying the data correlation of the nonbonded interaction and their impact on delG (**Fig 13**). In CXCR4+AMD3100 the dG_Bind_vdw has most significant impact on dG_Bind while dG_Bind_solv_GB has most negative impact on dG_Bind (**Fig 13A**). Similarly, all other cases, the correlation values significantly enumerating the impact of dG_Bind_vdw and dG_Bind_solv_GB has most negative impact on dG_Bind (**Fig 13**B, C, D).

## 4. Discussion

There is substantial evidence to support the hypothesis that some chemokines may restrict BC cell growth and metastasis, whereas others may stimulate angiogenesis and tumor progression in BC [48,49]. In breast cancer, an early loss of TGF-β growth control gives way to a basic dysregulation that mediates phenotypes and cell interactions that lead to invasive disease [50]. A meta-analysis showed that *CXCR4* is an efficient prognostic factor for breast cancer. Overexpression of *CXCR4* was significantly associated with lymph node status and distant metastasis and indicated poor overall and disease-free survival [51]. A study indicates that IL-19 increases tumor growth [52]. Research indicates that inhibition of BMP signalling may successfully target both the tumor and the surrounding microenvironment to reduce the tumor burden and metastasis [48]. The upregulation of VCAN is closely correlated with promoter methylation and the clinical features of breast cancer patients. Studies suggest that the breast cancer patient's survival rate gets lower when the VCAN

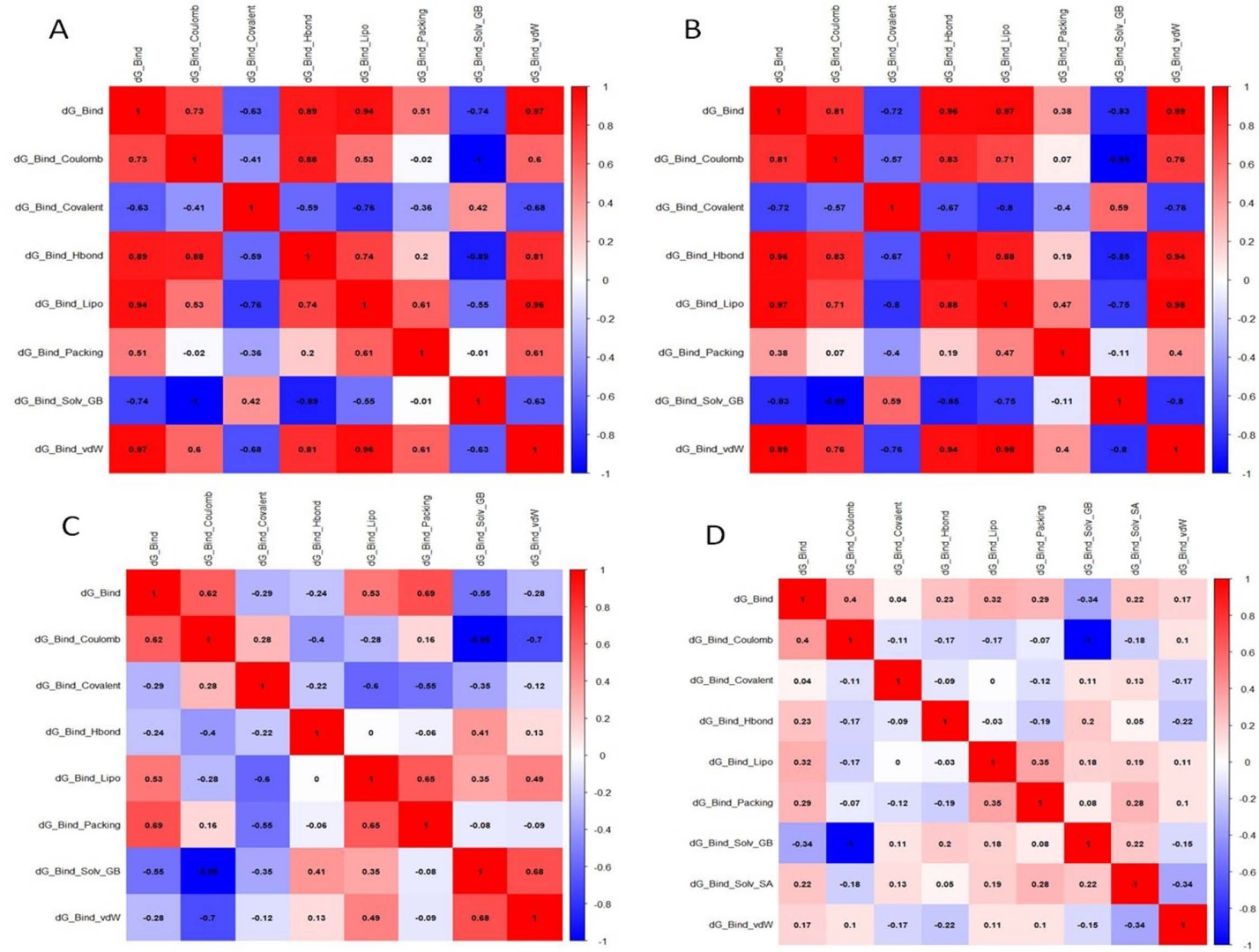

**Fig 13. Pearson correlation matrix exhibiting the relation of the weak intermolecular energies and their impact on delG_Bind (A) CXCR4+AMD3100, (B) CXCR4+D4476, (C) TGFβ-1+AMD3100, (D) TGFβ-1+D4476.**

expression level gets higher [34]. Studies have shown that Wnt signalling encompasses the proliferation, and the metastasis of breast cancer [53,54].

Several of the analyzed genes (TGFβ-1, IL19, CXCR4, BMP1, VCAN, and WNT2) showed elevated expression across multiple cancer types in pan-cancer datasets, including lung, colon, and ovarian cancers. This highlights their broader oncogenic roles beyond breast cancer, particularly in promoting tumor progression, metastasis, and immune modulation.

Such pan-cancer relevance reinforces their potential as universal therapeutic targets and supports the rationale for further functional validation across cancer types.

In this study, we elucidated through an *in silico* approach the involvement of the relevant genes coupled with their targeting aspect to better comprehend their role in breast cancer pathophysiology. In this regard, we have designed our study to work upon the expression analysis of the above-mentioned significant genes, and their specific role and

expression in pan-cancers including breast cancer. We also investigated their correlation with breast cancer metastasis, and their enrichment with the most significant pathways. The study further elaborated through the *in silico* targeting the simultaneous blockade of the *TGFβ-1* and *CXCR4* using their respective inhibitors (D4476 and AMD3100) in single as well as synergistic dimension (**Fig 14**).

The overexpression of TGF β-1, IL19, CXCR4, BMP1, VCAN, and WNT2 was seen across all PAN cancers. In comparison to non-tumor samples, it was observed that the protein levels of TGFβ-1, IL19, CXCR4, BMP1, VCAN, and WNT2 were significantly enhanced in several tumor samples (**Fig 1**). Our results revealed higher expression of the investigated genes in BRCA based on breast cancer subclass, individual cancer stages, and nodal metastasis status (**Fig 2**). We assessed the expression status of deregulated TGFβ-1, CXCR4, BMP1, VCAN, and WNT2 in the TISCH database, and the analysis reflected that TGFβ-1, CXCR4, BMP1, VCAN, and WNT2 are expressed higher in primary in relation to tumors that are metastatic in nature. Also, CXCR4 showed a significant increase in its expression, especially in the primary dataset (**Fig 3**). Using the Gene MANIA, the investigated genes showed significant interactions with *ADAM12* followed by *PAMR1, MMP11, VSTM4* and quite a few other genes as shown in **Fig 4**. This core network of *IL-19, TGFβ-1, CXCR4, BMP1, VCAN*, and *WNT2* reveals a tight-knit module that regulates tumor aggressiveness, immune evasion, and therapy resistance. Promoting tumor growth, migration, and M2 macrophage polarization, IL-19 has been shown to contribute to an immunosuppressive microenvironment in breast cancer and is a member of the IL-10 cytokine family. *TGFβ-1* plays a major role in epithelial-to-mesenchymal transition (EMT), immune suppression, and remodelling of the ECM, and is important for metastasis and resistance to therapy [55]. Tumor cell homing, metastasis, and maintenance of cancer stem cells are all significantly regulated by *CXCR4*, which acts in part through interaction with the *CXCL12* axis and co-opted in the signals from TGF β to increase invasiveness [49]. In turn, BMP1 activates latent TGF β-binding proteins and remodelling ECM proteins, to augment the invasive capabilities of tumor cells and modulate signalling by TGF beta families [56]. During invading and resistant tumors, they likely take part in the bright positive regulatory feedforward loops or cooperative signalling hubs-co-liberally with the circuits of *TGFβ–Wnt–CXCR4*-which sustain stemness, invasion, and immune escape. VCAN (versican), as large ECM proteoglycan, was related to immune exclusion through impairing T-cell infiltration and modulating inflammation, facilitating immune evasion and therapy resistance [57]. WNT2 supports cancer stemness, chemoresistance, and angiogenesis, and is co-activated in TGFβ-rich microenvironments in reports [58]. These genetic events might be linked with positive regulatory loops or combined signalling hubs-such as the

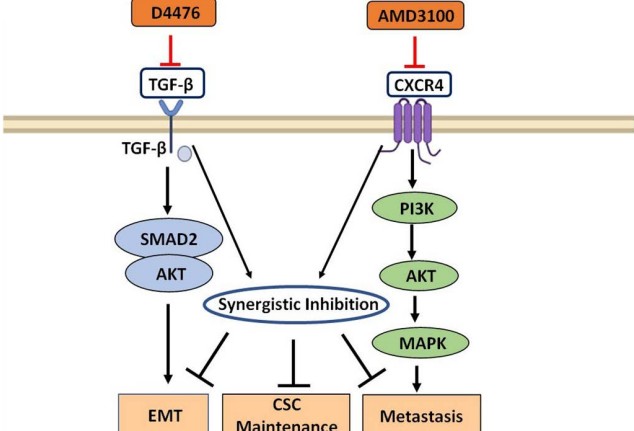

**Fig 14. Synergistic inhibition of TGF-β and CXCR4 pathways by D4476 and AMD3100.** The schematic shows how D4476 inhibits TGF-β signaling and AMD3100 blocks CXCR4, converging on shared downstream pathways (SMAD2, AKT, MAPK). This dual inhibition targets key cancer processes—EMT, CSC maintenance, and metastasis—proposing a synergistic therapeutic strategy in breast cancer.

*TGFβ-Wnt-CXCR4* circuits-that maintain tumor stemness, invasion, and immune escape. Their functional integration indicates that they are not only co-expressed but also potentially co-regulated, accentuating their translational value as combinatorial therapeutic targets. We further used DISCO database to find out the expression levels of investigated genes in different cells of the breast. The results revealed that *TGFβ-1, IL19, CXCR4, BMP1, VCAN,* and *WNT2* expression is enriched in several different breast tissues (**Fig 5**). The GO of investigated genes demonstrated that the genes are highly enriched in activities like Cytokine Activity, Receptor Ligand Activity, Metallopeptidase Activity, C-C Chemokine Binding, Type II Transforming Growth Factor Beta Receptor Binding, Type I Transforming Growth Factor Beta Receptor Binding, Chemokine Binding, Transforming Growth Factor Beta Receptor Binding, Chemokine Receptor Activity, positive Regulation of Glial Cell Differentiation, Regulation of Chemotaxis, and Response to Transforming Growth Factor Beta. Further analysis revealed that TGFβ-1, IL19, CXCR4, BMP1, VCAN, and WNT2 was abundant in Collagen-Containing Extracellular Matrix, Golgi Lumen, Perineuronal Net, and Perisynaptic Extracellular Matrix in terms GO analysis' CC terms. Also, the KEGG analysis revealed that the investigated genes were highly enriched in the pathways like cytokine-cytokine interaction, hepatocarcinoma, pathways in cancer, gastric cancer, etc. (**Fig 6**). The heat map showed the expression analysis of TGFβ-1, IL19, CXCR4, BMP1, VCAN, and WNT2. Among all the six genes, *TGFβ-1* showed high expression in primary tumors followed by *IL-19, BMP1*, and *CXCR4* (**Fig 7**). The expression pattern of TGFβ-1, IL19, CXCR4, BMP1, VCAN, and WNT2 in different molecular subclasses of Breast Cancer revealed that TGFβ-1, IL19, CXCR4, BMP1, VCAN, and WNT2 mRNA levels are higher in luminal A followed by luminal B, HER2 and Basal like respectively (**Fig 8**). The identification of the top 10 hub genes within the network of the investigated genes was accomplished through the utilisation of Cytohubba tool. *TGFβ-1, IL19, CXCR4, BMP1, VCAN, WNT2, POSTN, BGN, GDF5,* and *SDC4* were the genes included in the grid (**Fig 9**). Multiple ligand simultaneous docking (MLSD) of AMD3100 and D4476 with CXCR4 considering both the ligand docked at the same time at same position exhibited binding energy of −6.4 kcal/mol (**Fig 11C**). While MLSD of D4476 and AMD3100 with TGFβ-1 has showed comparable binding energy of −5.8 kcal/mol having polar contact with Ser1136 residue (**Fig 11F**). Therefore, from this docking study it can be signified that both AMD3100 and D4476 has higher propensity for CXCR4 and TGFβ-1 receptors and their consortium could invoke a significant level of inhibition to the target proteins. From the molecular Dynamic Simulation studies, it can be suggested that drugs bound to TGFβ-1are quite stable in complex due to higher affinity of the ligand (**Fig 12**). The results from the MM-GBSA suggested that the potential of AMD3100 molecule with TGFβ-1 and CXCR4, showed the efficiency in binding to the selected protein and the ability to form stable protein-ligand complexes (**Table 2**). Our findings in translational relevance open new avenues for using TGFβ-1 and CXCR4 as combinatorial therapeutic targets or prognostic markers, especially in the more aggressive subtypes of breast cancer. The most aggressive type of breast cancer is triple-negative breast cancer (TNBC), an entity defined by the absence of hormone receptors and HER2 expression [59]. It often shows high upregulation of both TGFβ-1 and CXCR4 levels, resulting in greater metastatic potential, chemoresistance, and poor patient survival [60]. In addition, these pathways may be co-targeted as a subtype-specific therapeutic strategy in cases where standard hormonal or HER2-targeted therapies fail. The applicability of our findings gains more pertinence in contemporary treatment alternatives. The standard-of-care treatments significantly improved the outcome of certain breast cancer subtypes; however, their efficacy remains limited in the drug-resistant and metastatic settings. While TNBCs do not respond to endocrine therapy and HER2-targeted therapies, immunotherapy has only limited benefit for a small fraction of patients [61]. The dual inhibition of TGFβ and CXCR4 is a combination of the inhibition of two distinct yet interrelated pathways involved in tumor progression, metastasis, and stemness. This therefore may even serve as an adjunct or alternative to conventional therapies in aggressive or relapsed disease settings, thus having promise for further translational investigation from preclinical to clinical settings. Therefore, our study provided significant insights into the expression patterns of TGFβ1 and CXCR4 and their involvement with other proteins, for instance VCAN, and WNT whose role is known in causing breast cancer stemness and hence cancer aggressiveness. Furthermore, we investigated two promising molecules AMD3100 and D4476 using synergistic approach for targeting TGFβ-1 and CXCR4 to be explored as better anti-cancer therapeutics

in future. The synergetic action of D4476 and AMD3100 can be attributed to their complementary action on two oncogenic axes. D4476 blocks TGFβ-1 signalling by inhibiting phosphorylation of SMAD2/3 and hence reduces programs of epithelial-mesenchymal transition and immunosuppressive transcription. AMD3100, meanwhile, blocks the CXCL12/ CXCR4 axis, preventing tumor cell migration and interaction with the stroma. The combined blockade probably destabilizes downstream converging pathways, such as PI3K/AKT and MAPK and WNT signalling, rendering it more efficient in anti-tumor activity. This mechanistic synergy provides for the therapeutic co-targeting of these axes in aggressive subtypes of breast cancer.

## 5. Conclusion and future prospects

Given the role of TGFβ-1 and CXCR4 in the breast cancer pathology especially their dysregulation in the said cancer, it is very important to develop better treatment interventions to address the same. In this direction, we employed a thorough bioinformatic approach, which in addition to revealing the dysregulation of TGFβ-1 and CXCR4 in various cancers also revealed that TGFβ-1 and CXCR4 when targeted using the respective inhibitors in combination displayed synergy, which is desirable for the better treatment of breast cancer. If this synergistic approach is replicated in further experiments as well especially in *in vitro* and *in vivo* experimentation after proper validation, it has the potential to enhance the existing treatment options available for curing breast cancer.

While simultaneously targeting TGF-β1 and CXCR4 suggests potential for treatment, resistance may develop through compensatory activation of parallel pathways, such as EGFR, Notch, or using alternative chemokine receptors (e.g., CCR7). Tumors may also up-regulate immune checkpoint molecules or rewire downstream effectors such as PI3K/AKT or MAPK to bypass inhibition. This emphasizes that such an approach should complement other targeted or immunotherapies and should be probed into in future studies.

Concerns about any potential off-target effects that may manifest as impaired wound healing, altered immune responses, or hemato-poietic dysfunction arise with inhibition of TGFβ-1 and CXCR4, which may be involved in the immune regulation, tissue homeostasis, and stem cell trafficking. Thus, therapeutic strategies targeting these pathways must consider selective targeting, localized delivery systems, or dosing schedules that balance safety and efficacy. Further preclinical studies are necessary to determine these risks in vivo.

Despite the valuable insights gained from bioinformatics analyses, our study acknowledges potential limitations, including the risk of false positives and overinterpretation due to dataset variability and algorithmic assumptions. These in silico predictions may not fully capture the biological complexity of tumor behavior in vivo. Therefore, our findings should be interpreted with caution and warrant further experimental validation to confirm their functional relevance in breast cancer.

## Author contributions

**Conceptualization:** Manzoor Ahmad Mir.

**Data curation:** Burhan Ul Haq, Shazia Sofi, Asma Jan, Mohammad Aljasir, Irshad Ahmad, Manzoor Ahmad Mir.

**Formal analysis:** Burhan Ul Haq, Shazia Sofi, Hina Qayoom, Nusrat Jan, Asma Jan, Mohammad Aljasir, Irshad Ahmad, Abdullah Almilabairy, Bader Alshehri, Fuzail Ahmad, Manzoor Ahmad Mir.

**Funding acquisition:** Irshad Ahmad, Manzoor Ahmad Mir.

**Investigation:** Burhan Ul Haq, Hina Qayoom, Nusrat Jan, Manzoor Ahmad Mir.

**Methodology:** Burhan Ul Haq, Shazia Sofi, Hina Qayoom, Manzoor Ahmad Mir.

**Project administration:** Manzoor Ahmad Mir.

**Resources:** Manzoor Ahmad Mir.

**Software:** Shazia Sofi.

**Supervision:** Burhan Ul Haq, Manzoor Ahmad Mir.

**Validation:** Burhan Ul Haq, Shazia Sofi, Irshad Ahmad, Abdullah Almilabairy, Bader Alshehri, Fuzail Ahmad.

**Visualization:** Shazia Sofi, Hina Qayoom, Nusrat Jan, Asma Jan, Mohammad Aljasir, Irshad Ahmad, Abdullah Almilabairy, Fuzail Ahmad, Manzoor Ahmad Mir.

**Writing – original draft:** Burhan Ul Haq, Shazia Sofi.

**Writing – review & editing:** Shazia Sofi, Hina Qayoom, Nusrat Jan, Asma Jan, Mohammad Aljasir, Irshad Ahmad, Abdullah Almilabairy, Bader Alshehri, Fuzail Ahmad, Manzoor Ahmad Mir.

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
