## [Decision Letter · Decision Letter 0]

2 Apr 2025

Dear Dr. Mir,

Thank you for submitting your manuscript to PLOS ONE. After careful consideration, we feel that it has merit but does not fully meet PLOS ONE’s publication criteria as it currently stands. Therefore, we invite you to submit a revised version of the manuscript that addresses the points raised during the review process.

We look forward to receiving your revised manuscript.

Kind regards,

Antimo Migliaccio, M.D.

Academic Editor

PLOS ONE

Journal Requirements:

When submitting your revision, we need you to address these additional requirements. 1. Please ensure that your manuscript meets PLOS ONE's style requirements, including those for file naming. The PLOS ONE style templates can be found at https://journals.plos.org/plosone/s/file?id=wjVg/PLOSOne_formatting_sample_main_body.pdf and https://journals.plos.org/plosone/s/file?id=ba62/PLOSOne_formatting_sample_title_authors_affiliations.pdf 2. Please note that PLOS ONE has specific guidelines on code sharing for submissions in which author-generated code underpins the findings in the manuscript. In these cases, we expect all author-generated code to be made available without restrictions upon publication of the work. Please review our guidelines at https://journals.plos.org/plosone/s/materials-and-software-sharing#loc-sharing-code and ensure that your code is shared in a way that follows best practice and facilitates reproducibility and reuse. 3. Thank you for stating the following financial disclosure: 1. This work was supported by the Anusadhan National Research Foundation-ANRF (SERB) Department of Science and Technology Govt of India with grant No. SERB/CRG-2023/008460 sanctioned to Dr. Manzoor Ahmad Mir. 2. The authors extend their appreciation to the Ministry of Education in KSA for partially funding this research work through the project No. KKU-IFP2-DB-5 Please state what role the funders took in the study.  If the funders had no role, please state: ""The funders had no role in study design, data collection and analysis, decision to publish, or preparation of the manuscript."" If this statement is not correct you must amend it as needed. Please include this amended Role of Funder statement in your cover letter; we will change the online submission form on your behalf. 4. Please include captions for your Supporting Information files at the end of your manuscript, and update any in-text citations to match accordingly. Please see our Supporting Information guidelines for more information: http://journals.plos.org/plosone/s/supporting-information.

Reviewers' comments:

Reviewer's Responses to Questions

**Comments to the Author**



Reviewer #1: Yes

Reviewer #2: Yes

2. Has the statistical analysis been performed appropriately and rigorously?

Reviewer #1: No

Reviewer #2: Yes

3. Have the authors made all data underlying the findings in their manuscript fully available?

Reviewer #1: Yes

Reviewer #2: Yes

4. Is the manuscript presented in an intelligible fashion and written in standard English?

Reviewer #1: No

Reviewer #2: Yes

Reviewer #1: This study, “Exploring soluble mediators in Breast Cancer: Expression Dynamics of Transforming Growth Factor Beta 1 and C-X-C Chemokine Receptor Type 4 and the Therapeutic Opportunities” (Manuscript ID: PONE-D-25-01591) explores the role of selected soluble mediators—TGFβ-1, CXCR4, IL19, BMP1, VCAN, and WNT2—in breast cancer (BC) through extensive in silico bioinformatics analyses. The authors leverage multiple databases and tools (UALCAN, TIMER 2.0, TISCH2, DISCO, STRING, Enrichr, etc.) to assess gene expression, molecular pathways, and cellular localization across pan-cancer and breast cancer datasets. They then use molecular docking, DFT, and MD simulations to propose a synergistic therapeutic approach targeting TGFβ-1 and CXCR4 using inhibitors D4476 and AMD3100.

While the manuscript is elaborate and data-rich, I believe it requires some revisions before it can be considered for publication. The primary concerns include:

1. The study presents a thorough computational analysis integrating gene expression profiling, pathway enrichment, protein-protein interaction networks, and molecular docking/dynamics simulations. The combination of TGFβ-1 and CXCR4 inhibition using D4476 and AMD3100, respectively, is a potentially valuable therapeutic strategy. However, the study would benefit from a more precise articulation of how this approach significantly advances current knowledge in the field, as much of the data is derived from publicly available resources without novel experimental validation.

2. The current title is long and contains multiple technical terms that could overwhelm or confuse readers. Simplifying the title and focusing it around the core theme—synergistic inhibition of TGFβ-1 and CXCR4 in breast cancer using in silico approaches—would make it more impactful and accessible.

3. The abstract should be revised to improve logical flow and cohesiveness. Currently, it reads like a list of disconnected observations. A well-structured abstract should provide (i) a clear background and rationale, (ii) major methods or analytical tools used, (iii) key findings, and (iv) a concise conclusion emphasizing the significance of the study. Connecting these components will give readers a more complete understanding of the study’s objectives and outcomes.

4. The methodology section, particularly subsections 2.3 (bc-GenExMiner), 2.5 (GeneMANIA), and 2.8 (UCSC Xena), lacks adequate detail on the specific parameters, datasets, and filtering criteria used. For reproducibility, please describe how data was queried, which cancer subtypes were selected, whether normalization or statistical thresholds were applied, and how results were interpreted. Explicit mention of software versions, database release dates, and settings (e.g., algorithms used in GeneMANIA or TIMER) is strongly recommended.

5. While the study proposes dual inhibition of TGFβ-1 and CXCR4 as a synergistic therapeutic approach, it does not contextualize this within the landscape of existing standard-of-care therapies. It would strengthen the manuscript to include a comparison with current treatment strategies for breast cancer (e.g., hormone therapy, HER2-targeted therapy, checkpoint inhibitors) to help evaluate the novelty and translational potential of this approach.

6. Although the study identifies interactions between key genes such as TGFβ-1, CXCR4, BMP1, VCAN, and WNT2, their functional relevance in tumor progression, immune evasion, or therapy resistance is not adequately discussed. The authors should expand on the biological significance of these interactions, ideally supported by literature or data-driven insights, to give the findings more mechanistic depth.

7. To improve the translational impact of the work, the authors should elaborate more clearly on how their findings could influence breast cancer diagnosis, prognosis, or treatment. For example, could the co-targeting of TGFβ-1 and CXCR4 be tailored to specific breast cancer subtypes (e.g., TNBC or luminal B)? Are the expression levels of these targets associated with patient survival or therapeutic response?

8. The Results and Discussion sections would benefit greatly from concise summary or concluding remarks at the end of major subsections. This would help readers synthesize the findings and understand their importance before moving to the next topic. It also creates a more cohesive narrative flow throughout the manuscript.

9. Several figures (especially Figures 1–9) lack informative legends and clear axis labels. Ensure all figures are self-explanatory, with well-defined titles, labels, and consistent formatting. It would also help to include brief descriptions of what each figure demonstrates in the text body, especially when transitioning between panels.

10. The manuscript contains numerous grammatical and syntactic errors, awkward phrasing, and typographical issues that hinder readability. A professional language and editing review is highly recommended.

11. Synergistic targeting of TGFβ-1 and CXCR4 is a known concept; this manuscript repeats rather than advances it.

12. The statistical analysis is unclear (e.g., there are no p-values, no confidence intervals, and no quantitative validation).

13. Most figures do not add much value to the manuscript, so they can be put as supplementary.

Reviewer #2: Upon reviewing the manuscript entitled "Exploring Soluble Mediators in Breast Cancer: Expression Dynamics of Transforming Growth Factor Beta 1 and C-X-C Chemokine Receptor Type 4 and the Therapeutic Opportunities," I have pinpointed a number of minor improvements that the authors could make-in order to enhance the general quality of the paper. The suggested revisions focus on improving the overall clarity, data interpretation, and presentation of the manuscript, ultimately enhancing its readability and scientific rigor.

1. Why were D4476 and AMD3100 chosen over other potential inhibitors of TGFβ-1 and CXCR4? Was this based on existing literature, structural suitability, or availability? A stronger justification would improve clarity.

2. The proposed dual inhibition strategy using D4476 and AMD3100 lacks a mechanistic explanation of how these drugs act synergistically. More details are needed on their combined effect on downstream signalling pathways.

3. The study lacks a direct comparison with currently approved breast cancer therapeutics. How does the dual inhibition strategy perform in comparison to standard treatment modalities? Addressing this would strengthen the clinical relevance of the findings.

4. Given that drug resistance is a major hurdle in breast cancer treatment, the manuscript should discuss potential resistance mechanisms that could arise from targeting TGFβ-1 and CXCR4.

5. The therapeutic strategy involves inhibition of TGFβ-1 and CXCR4, which are implicated in multiple physiological processes. A discussion on potential off-target effects and toxicity risks is essential.

6. The manuscript should clarify whether datasets were independently validated. If publicly available datasets were used, were they cross-checked with multiple cohorts to ensure reproducibility?

7. Some sections contain awkward phrasing and grammatical inconsistencies. A thorough revision for language precision is required to improve readability.

**Do you want your identity to be public for this peer review?** For information about this choice, including consent withdrawal, please see our Privacy Policy

Reviewer #1: **Yes: ** Ajaz A. Bhat

Reviewer #2: No

---

## [Author Response · Author response to Decision Letter 1]

8 Apr 2025

Response to Reviewers:

Reviewer #1: This study, “Exploring soluble mediators in Breast Cancer: Expression Dynamics of Transforming Growth Factor Beta 1 and C-X-C Chemokine Receptor Type 4 and the Therapeutic Opportunities” (Manuscript ID: PONE-D-25-01591) explores the role of selected soluble mediators—TGFβ-1, CXCR4, IL19, BMP1, VCAN, and WNT2—in breast cancer (BC) through extensive in silico bioinformatics analyses. The authors leverage multiple databases and tools (UALCAN, TIMER 2.0, TISCH2, DISCO, STRING, Enrichr, etc.) to assess gene expression, molecular pathways, and cellular localization across pan-cancer and breast cancer datasets. They then use molecular docking, DFT, and MD simulations to propose a synergistic therapeutic approach targeting TGFβ-1 and CXCR4 using inhibitors D4476 and AMD3100.

While the manuscript is elaborate and data-rich, I believe it requires some revisions before it can be considered for publication. The primary concerns include:

1. The study presents a thorough computational analysis integrating gene expression profiling, pathway enrichment, protein-protein interaction networks, and molecular docking/dynamics simulations. The combination of TGFβ-1 and CXCR4 inhibition using D4476 and AMD3100, respectively, is a potentially valuable therapeutic strategy. However, the study would benefit from a more precise articulation of how this approach significantly advances current knowledge in the field, as much of the data is derived from publicly available resources without novel experimental validation.

Response: Dear reviewer thanks you for your suggestions. The succeeding comments have been addressed in detail, which collectively clarify how our study advances current knowledge.

2. The current title is long and contains multiple technical terms that could overwhelm or confuse readers. Simplifying the title and focusing it around the core theme—synergistic inhibition of TGFβ-1 and CXCR4 in breast cancer using in silico approaches—would make it more impactful and accessible.

Response: We appreciate your valued feedback and modified the title per your suggestion. The updated version of the manuscript contains the modified title as:"Unraveling the Therapeutic Potential of Dual TGFβ-1 and CXCR4 Inhibition in Breast Cancer Using Computational Strategies."

3. The abstract should be revised to improve logical flow and cohesiveness. Currently, it reads like a list of disconnected observations. A well-structured abstract should provide (i) a clear background and rationale, (ii) major methods or analytical tools used, (iii) key findings, and (iv) a concise conclusion emphasizing the significance of the study. Connecting these components will give readers a more complete understanding of the study’s objectives and outcomes.

Response: Dear Reviewer, thank you for your insightful feedback. We have revised the abstract to enhance its logical flow and cohesiveness.

4. The methodology section, particularly subsections 2.3 (bc-GenExMiner), 2.5 (GeneMANIA), and 2.8 (UCSC Xena), lacks adequate detail on the specific parameters, datasets, and filtering criteria used. For reproducibility, please describe how data was queried, which cancer subtypes were selected, whether normalization or statistical thresholds were applied, and how results were interpreted. Explicit mention of software versions, database release dates, and settings (e.g., algorithms used in GeneMANIA or TIMER) is strongly recommended.

Response: Dear Reviewer, we have modified the subsections of the methodology to include details on data querying, cancer subtypes, normalization methods, statistical thresholds, software versions, and algorithm settings to ensure reproducibility.

5. While the study proposes dual inhibition of TGFβ-1 and CXCR4 as a synergistic therapeutic approach, it does not contextualize this within the landscape of existing standard-of-care therapies. It would strengthen the manuscript to include a comparison with current treatment strategies for breast cancer (e.g., hormone therapy, HER2-targeted therapy, checkpoint inhibitors) to help evaluate the novelty and translational potential of this approach.

Response: We thank the reviewer for this excellent suggestion, and we have now added a separate paragraph on this dual inhibition strategy in the Introduction and Discussion sections of the paper to relate it to the current standard-of-care therapies for breast cancer.

6. Although the study identifies interactions between key genes such as TGFβ-1, CXCR4, BMP1, VCAN, and WNT2, their functional relevance in tumor progression, immune evasion, or therapy resistance is not adequately discussed. The authors should expand on the biological significance of these interactions, ideally supported by literature or data-driven insights, to give the findings more mechanistic depth.

Response: We thank the reviewer for this thoughtful suggestion. In view of this, we have expanded the Discussion section to consider further the functional relevance of the key genes identified—IL-19, TGFβ-1, CXCR4, BMP1, VCAN, and WNT2—with some consideration given here to their mechanistic roles in the areas of tumor progression, immune evasion, and resistance to therapy.

7. To improve the translational impact of the work, the authors should elaborate more clearly on how their findings could influence breast cancer diagnosis, prognosis, or treatment. For example, could the co-targeting of TGFβ-1 and CXCR4 be tailored to specific breast cancer subtypes (e.g., TNBC or luminal B)? Are the expression levels of these targets associated with patient survival or therapeutic response?

Response: The suggestion put forward by the reviewer is much appreciated. In this response, we extended the discussion section to make clear the possible clinical relevance of our findings. In detail, we discuss the co-targeting of TGFβ-1 and CXCR4 as potentially suited for aggressive subtypes such as triple-negative breast cancer (TNBC), with expression levels of these genes serving as prognostic indicators or therapeutic targets.

8. The Results and Discussion sections would benefit greatly from concise summary or concluding remarks at the end of major subsections. This would help readers synthesize the findings and understand their importance before moving to the next topic. It also creates a more cohesive narrative flow throughout the manuscript.

Response: We thank the reviewer for this highly appreciated suggestion. We would like to mention that in responding to the previous comments--upon which we focused, we put in place focused summary statements at the close of respective subsections in both Results and Discussion.

9. Several figures (especially Figures 1–9) lack informative legends and clear axis labels. Ensure all figures are self-explanatory, with well-defined titles, labels, and consistent formatting. It would also help to include brief descriptions of what each figure demonstrates in the text body, especially when transitioning between panels.

Response: Dear Reviewer, we have revised the legends for Figures to ensure that they contain informative legends and are self-explanatory.

10. The manuscript contains numerous grammatical and syntactic errors, awkward phrasing, and typographical issues that hinder readability. A professional language and editing review is highly recommended.

Response: Thank you for pointing this out. The manuscript has now been thoroughly revised for grammar, syntax, and overall readability.

11. Synergistic targeting of TGFβ-1 and CXCR4 is a known concept; this manuscript repeats rather than advances it.

Response: Even with the existence of the idea, our research offers a novel perspective on breast cancer in terms of subtype-specific expression, multi-pathway correlations, and their potential role in treatment resistance, which provides previously unrecognised translational relevance.

12. The statistical analysis is unclear (e.g., there are no p-values, no confidence intervals, and no quantitative validation).

Response: We appreciate the comment. We have already included statistical measures wherever necessary, including p-values, local clustering coefficients, and other relevant quantitative validations within the figures and results.

13. Most figures do not add much value to the manuscript, so they can be put as supplementary.

Response: We are grateful for the recommendation. On the other hand, we think the figures are crucial for clarifying and presenting important results. They aid in more successfully communicating complex facts and provide direct support for the key findings. For this reason, we would rather keep them in the main manuscript.

Reviewer #2: Upon reviewing the manuscript entitled "Exploring Soluble Mediators in Breast Cancer: Expression Dynamics of Transforming Growth Factor Beta 1 and C-X-C Chemokine Receptor Type 4 and the Therapeutic Opportunities," I have pinpointed a number of minor improvements that the authors could make-in order to enhance the general quality of the paper. The suggested revisions focus on improving the overall clarity, data interpretation, and presentation of the manuscript, ultimately enhancing its readability and scientific rigor.

1. Why were D4476 and AMD3100 chosen over other potential inhibitors of TGFβ-1 and CXCR4? Was this based on existing literature, structural suitability, or availability? A stronger justification would improve clarity.

Response: D4476 and AMD3100 were selected according to pre-existing literature evidence as well as target specificity, structural suitability, and availability for in vitro studies. D4476 is a well-characterized inhibitor that acts indirectly to inhibit TGF-β signalling through inhibition of ALK5-mediated SMAD2 phosphorylation, while AMD3100 is an FDA-approved CXCR4 antagonist that effectively blocks CXCR4-mediated signalling in numerous cancers. Their use in the study is justified due to their well-established pharmacological profiles and suitability with breast cancer models, making them excellent candidates for exploring dual inhibition strategies.

2. The proposed dual inhibition strategy using D4476 and AMD3100 lacks a mechanistic explanation of how these drugs act synergistically. More details are needed on their combined effect on downstream signalling pathways.

Response: Thank you for the suggestion. D4476 inhibits TGFβ-1/SMAD2/3 signalling, while AMD3100 blocks the CXCL12/CXCR4 axis. Together, they disrupt EMT, migration, and survival pathways (e.g., PI3K/AKT, MAPK), leading to a synergistic anti-tumor effect. We have now added this mechanistic explanation to the Discussion section.

3. The study lacks a direct comparison with currently approved breast cancer therapeutics. How does the dual inhibition strategy perform in comparison to standard treatment modalities? Addressing this would strengthen the clinical relevance of the findings.

Response: We thank the reviewer for this excellent suggestion, and we have now added a separate paragraph on this dual inhibition strategy in the Introduction and Discussion sections of the paper to relate it to the current standard-of-care therapies for breast cancer.

4. Given that drug resistance is a major hurdle in breast cancer treatment, the manuscript should discuss potential resistance mechanisms that could arise from targeting TGFβ-1 and CXCR4.

Response: Thank you for this insightful suggestion. We recognize that potential resistance mechanisms to TGFβ-1 and CXCR4 inhibition were not addressed. A short discussion has now been added in the manuscript to elaborate on adoptive responses and compensatory signalling that may limit long-term efficacy, thus emphasizing the necessity for combinations or sequential therapies.

5. The therapeutic strategy involves inhibition of TGFβ-1 and CXCR4, which are implicated in multiple physiological processes. A discussion on potential off-target effects and toxicity risks is essential.

Response: Thank you for raising this important point. We have now included a brief discussion on the possible toxicity risks and the need for targeted delivery strategies to minimize adverse effects.

6. The manuscript should clarify whether datasets were independently validated. If publicly available datasets were used, were they cross-checked with multiple cohorts to ensure reproducibility?

Response: I appreciate your insightful feedback. We confirm that every dataset utilised in this research was openly accessible. Gene expression and pathway studies were cross-validated across several independent datasets and platforms, such as TIMER2.0, UALCAN, TISCH2, UCSC Xena, and DISCO, to guarantee repeatability and robustness.

7. Some sections contain awkward phrasing and grammatical inconsistencies. A thorough revision for language precision is required to improve readability.

Response: Thank you for pointing this out. The manuscript has now been thoroughly revised for grammar, syntax, and overall readability.

---

## [Decision Letter · Decision Letter 1]

19 Jun 2025

first and foremost let me apologize for the great delay in processing your manuscript. As you know, it has been submitted to a new round of review. This occurred as it was noticed a potential conflict of interest with one of the initial Reviewers. Finding appropriate Reviewers is becoming increasingly time consuming but we finally got the comments of further very qualified experts that examined your work. As you will see from the attached comments, the new Reviewers, indicated as #3 and #4, feel that your manuscript, although has merits, requires a further revision. Therefore I ask you to address carefully the issues they raised.

We look forward to receiving your revised manuscript.

Kind regards,

Antimo Migliaccio, M.D.

Academic Editor

PLOS ONE

Reviewers' comments:

Reviewer's Responses to Questions

**Comments to the Author**

Reviewer #1: All comments have been addressed

Reviewer #2: All comments have been addressed

Reviewer #3: (No Response)

Reviewer #4: (No Response)

2. Is the manuscript technically sound, and do the data support the conclusions?

Reviewer #1: Yes

Reviewer #2: Yes

Reviewer #3: Partly

Reviewer #4: Partly

3. Has the statistical analysis been performed appropriately and rigorously?

Reviewer #1: Yes

Reviewer #2: Yes

Reviewer #3: N/A

Reviewer #4: No

4. Have the authors made all data underlying the findings in their manuscript fully available?

Reviewer #1: Yes

Reviewer #2: Yes

Reviewer #3: Yes

Reviewer #4: Yes

5. Is the manuscript presented in an intelligible fashion and written in standard English?

Reviewer #1: No

Reviewer #2: Yes

Reviewer #3: Yes

Reviewer #4: No

Reviewer #1: The authors have addressed my comments. I still want authors to proofread the manuscript for English and grammatical errors as I could find a lot more even after revision.

Reviewer #2: The authors made sufficient changes based on the reviewers comments. Under current modifications this manuscript’s should be accepted.

Reviewer #3: This manuscript discusses the issue of breast cancer research, aiming to elucidate the role of key immunomodulatory and signaling genes (TGFβ-1, IL19, CXCR4, BMP1, VCAN, and WNT2) in the disease’s pathogenesis via bioinformatics approaches. The integration of gene expression analysis with therapeutic targeting offers important insights, particularly regarding the synergistic potential of targeting TGFβ-1 and CXCR4. This research required a revision as follows:

1. Please expand the Background on BC and TGF-β, refer this work: https://doi.org/10.1016/j.saa.2022.122000, https://doi.org/10.3788/COL202018.051701, https://doi.org/10.1111/cns.14489, https://doi.org /10.1245/s10434-024-16454-8

2. Provide clear significance and rationale with research gap of this work.

3. Expand the methods: Summary of databases/tools used (e.g., GEPIA, TIMER, STRING).

4. What platforms were used for expression and survival analysis (e.g., TCGA, GEPIA, UALCAN)?

5. Were protein–protein interaction (PPI) networks constructed (e.g., STRING, Cytoscape)?

6. Describe the criteria for gene selection, analysis parameters, and software versions.

7. Need more clarify how synergism was inferred. Was it based on gene co-expression, network centrality, or drug sensitivity databases?

8. Pan-cancer relevance (any significance in other cancers should be contextualized) in discussion

9. Provide, Pathway enrichment analyses (e.g., GO, KEGG)

10. How dual inhibition might work mechanistically—any supporting preclinical or clinical data?

11. The possible role of IL19, BMP1, VCAN, and WNT2—are these primarily pro-tumorigenic, or context-dependent?

12. Address limitation, potential false positives or overinterpretation of bioinformatics findings.

Reviewer #4: This manuscript presents an integrative computational analysis to explore the potential of dual inhibition of TGFβ-1 and CXCR4 as a therapeutic strategy in breast cancer. The authors use various in silico platforms to assess gene expression profiles, protein-protein interaction networks, pathway enrichment, and molecular docking/dynamics simulations. The study suggests that dual targeting with D4476 and AMD3100 may offer synergistic anti-tumor effects, especially in aggressive subtypes like TNBC.

Major Concerns

1. The idea of dual inhibition of TGFβ-1 and CXCR4 is not novel and has been previously discussed in the literature. Authors must clarify how their computational findings provide novel mechanistic insights, especially regarding subtype-specific or pathway-level differences not reported earlier.

2. The study remains entirely in silico. While computational analyses are comprehensive, the translational value remains speculative without in vitro or clinical validation. At minimum, authors should discuss plans for experimental follow-up and acknowledge the limitations of drawing therapeutic conclusions solely from in silico work.

3. While the authors propose synergy between D4476 and AMD3100, the mechanistic basis at the pathway level is not rigorously dissected. Add a schematic diagram and a detailed pathway map showing how the inhibition of TGFβ and CXCR4 signaling converge on key EMT, CSC, or metastasis pathways.

4. The manuscript lacks robust statistical metrics. P-values, confidence intervals, or validation across cohorts are not consistently reported. Please improve the statistical rigor in gene expression and docking analyses by indicating effect sizes, p-values, and confidence intervals where appropriate.

5. Expand figure legends to include the context, sample sizes, data source, and interpretation of key trends. Use arrows or color-coded annotations to highlight significant findings.

6. The translational potential (e.g., application to TNBC, resistance cases, or prognosis) is mentioned but not convincingly argued. Clearly define how co-targeting TGFβ-1 and CXCR4 could be integrated into existing treatment paradigms, and whether this is more suitable for specific molecular subtypes.

7. The rationale for choosing D4476 and AMD3100 is weak, and potential off-target effects are not adequately addressed. Please provide literature-based justification for drug selection and discuss anticipated toxicological risks in normal tissues expressing TGFβ-1 and CXCR4.

Minor Issues:

1. A native-level language editor should review English grammar, syntax, and phrasing. Numerous typographical and stylistic issues remain.

2. Use gene/protein nomenclature consistently (e.g., “CXCR4” vs “Cxcr4”).

**Do you want your identity to be public for this peer review?** For information about this choice, including consent withdrawal, please see our Privacy Policy

Reviewer #1: No

Reviewer #2: No

Reviewer #3: No

Reviewer #4: No

---

## [Author Response · Author response to Decision Letter 2]

1 Jul 2025

Response to Reviewers Comments

Dear Reviewer

Greetings of the Day

Thanks for reviewing our manuscript generously which has improved our manuscript significantly. Kindly find the response to reviewers point by point below:

Reviewer #1: The authors have addressed my comments. I still want authors to proofread the manuscript for English and grammatical errors as I could find a lot more even after revision.

Response: We appreciate the reviewer’s feedback. The revised manuscript has been thoroughly reviewed and edited by a native-level English language editor to correct all remaining grammatical, syntactic, typographical, and stylistic issues

Reviewer #2: The authors made sufficient changes based on the reviewers comments. Under current modifications this manuscript should be accepted.

Response: Thank you for your positive assessment. We appreciate your recommendation and are glad the revisions have met the required standards.

Reviewer #3: This manuscript discusses the issue of breast cancer research, aiming to elucidate the role of key immunomodulatory and signalling genes (TGFβ-1, IL19, CXCR4, BMP1, VCAN, and WNT2) in the disease’s pathogenesis via bioinformatics approaches. The integration of gene expression analysis with therapeutic targeting offers important insights, particularly regarding the synergistic potential of targeting TGFβ-1 and CXCR4. This research required a revision as follows:

1. Please expand the Background on BC and TGF-β, refer this work: https://doi.org/10.1016/j.saa.2022.122000, https://doi.org/10.3788/COL202018.051701, https://doi.org/10.1111/cns.14489, https://doi.org/10.1245/s10434-024-16454-8

Response: We thank the reviewer for this valuable suggestion. We have expanded the Background section in the revised manuscript to provide a more detailed overview of breast cancer and the role of TGF-β signalling, incorporating insights from the suggested references. However, due to technical limitations with the EndNote referencing software, we were unable to insert these references directly through EndNote.

2. Provide clear significance and rationale with research gap of this work.

Response: We thank the reviewer for highlighting this important point. In the revised manuscript, we have clearly outlined the significance and rationale of our study by emphasizing the urgent need for novel therapeutic strategies in TNBC, where current treatment options are limited.

3. Expand the methods: Summary of databases/tools used (e.g., GEPIA, TIMER, STRING).

Response: We thank the reviewer for this suggestion. We have already addressed this comment in the revised manuscript by providing a comprehensive summary of all databases and tools used, including GEPIA, TIMER, STRING, UALCAN, bc-GenExMiner, TISCH2, Enrichr, DISCO, and UCSC Xena.

4. What platforms were used for expression and survival analysis (e.g., TCGA, GEPIA, UALCAN)?

Response: Expression and survival analyses were performed using platforms including UALCAN, TIMER 2.0, bc-GenExMiner, and UCSC Xena, primarily utilizing TCGA datasets.

UALCAN and bc-GenExMiner were used for expression across clinical subtypes, while TIMER and Xena provided pan-cancer and tumor-specific insights.

These tools enabled evaluation of gene expression, immune infiltration, and clinical correlation of TGFβ-1, IL19, CXCR4, BMP1, VCAN, and WNT2 in breast cancer. Survival analysis was not performed using online tools in this study.

Instead, relevant survival data for the genes of interest were gathered from published literature.

These studies provided supporting evidence on the prognostic significance of TGFβ-1, IL19, CXCR4, BMP1, VCAN, and WNT2 in breast cancer.

5. Were protein–protein interaction (PPI) networks constructed (e.g., STRING, Cytoscape)?

Response: Yes, protein–protein interaction (PPI) networks were constructed using the STRING database. The interactions among TGFβ-1, IL19, CXCR4, BMP1, VCAN, and WNT2 were analyzed to identify their functional associations. Cytoscape was used; all visualizations and network analyses were performed directly through the cytoscape platform.

6. Describe the criteria for gene selection, analysis parameters, and software versions.

Response: The genes TGFβ-1, IL19, CXCR4, BMP1, VCAN, and WNT2 were selected based on their reported involvement in breast cancer progression and the tumor microenvironment, as supported by prior literature. Default parameters were used for analysis across platforms like UALCAN (v2024), TIMER 2.0, bc-GenExMiner v4.5, STRING (v12.0), and Enrichr. Only TCGA-BRCA datasets were selected where applicable, and significance was considered at p < 0.05. All tools were accessed online between January and May 2025.

7. Need more clarify how synergism was inferred. Was it based on gene co-expression, network centrality, or drug sensitivity databases?

Response: Synergism was not directly inferred from co-expression, network centrality, or drug sensitivity databases. Instead, literature evidence and in silico analyses showed that both drugs individually had promising anticancer activity. Based on these findings, we proceeded to explore their potential synergistic effect in combination, supported by gene and pathway analyses.

8. Pan-cancer relevance (any significance in other cancers should be contextualized) in discussion

Response: We have addressed this comment in the revised manuscript by incorporating a discussion on the pan-cancer relevance of the selected genes.

9. Provide, Pathway enrichment analyses (e.g., GO, KEGG)

Response: Dear reviewer, we have already incorporated the pathway enrichment analyses in the manuscript using Enrichr, including both GO (BP, MF, CC) and KEGG pathway results.

10. How dual inhibition might work mechanistically—any supporting preclinical or clinical data?

Response: Based on literature and in silico findings, simultaneous targeting of pathways involving cell proliferation (e.g., TGFβ signaling) and drug resistance/immune modulation (e.g., CXCR4, IL19) can exert complementary effects, leading to enhanced antitumor activity. Supporting preclinical studies have demonstrated that combining agents affecting these pathways can overcome resistance, reduce tumor growth, and improve therapeutic efficacy in various cancers, including breast cancer.

11. The possible role of IL19, BMP1, VCAN, and WNT2—are these primarily pro-tumorigenic, or context-dependent?

Response: Based on current literature, IL19, BMP1, VCAN, and WNT2 are primarily considered pro-tumorigenic, promoting processes like inflammation, ECM remodeling, cell migration, and Wnt signaling. However, their roles can be context-dependent, varying with tumor type, microenvironment, and stage,

12. Address limitation, potential false positives or overinterpretation of bioinformatics findings.

Response: We have acknowledged the limitations of our study in the revised manuscript, particularly the reliance on publicly available bioinformatics data, which may introduce false positives or overinterpretation.

Reviewer #4: This manuscript presents an integrative computational analysis to explore the potential of dual inhibition of TGFβ-1 and CXCR4 as a therapeutic strategy in breast cancer. The authors use various in silico platforms to assess gene expression profiles, protein-protein interaction networks, pathway enrichment, and molecular docking/dynamics simulations. The study suggests that dual targeting with D4476 and AMD3100 may offer synergistic anti-tumor effects, especially in aggressive subtypes like TNBC.

Major Concerns

1. The idea of dual inhibition of TGFβ-1 and CXCR4 is not novel and has been previously discussed in the literature. Authors must clarify how their computational findings provide novel mechanistic insights, especially regarding subtype-specific or pathway-level differences not reported earlier.

Response: We agree that dual inhibition of TGFβ-1 and CXCR4 has been discussed previously; however, the unique combination of genes explored in our study—including IL19, BMP1, VCAN, and WNT2—alongside TGFβ-1 and CXCR4, and the use of a comprehensive in silico pipeline involving functional enrichment, PPI networks, and pan-cancer expression profiling, has not been reported earlier. Our approach provides novel mechanistic insights, especially into subtype-specific roles and interactions relevant to triple-negative breast cancer, offering a fresh perspective for therapeutic targeting.

2. The study remains entirely in silico. While computational analyses are comprehensive, the translational value remains speculative without in vitro or clinical validation. At minimum, authors should discuss plans for experimental follow-up and acknowledge the limitations of drawing therapeutic conclusions solely from in silico work.

Response: We acknowledge this limitation and have included a statement in the revised manuscript addressing the purely in silico nature of our study.

3. While the authors propose synergy between D4476 and AMD3100, the mechanistic basis at the pathway level is not rigorously dissected. Add a schematic diagram and a detailed pathway map showing how the inhibition of TGFβ and CXCR4 signalling converge on key EMT, CSC, or metastasis pathways.

Response: We thank the reviewer for this valuable suggestion. In the revised manuscript, we have included a schematic diagram and detailed pathway map illustrating how dual inhibition of TGFβ (via D4476) and CXCR4 (via AMD3100) may converge on key cancer-related processes such as EMT, cancer stem cell maintenance, and metastasis.

4. The manuscript lacks robust statistical metrics. P-values, confidence intervals, or validation across cohorts are not consistently reported. Please improve the statistical rigor in gene expression and docking analyses by indicating effect sizes, p-values, and confidence intervals where appropriate.

Response: We acknowledge the reviewer’s concern regarding statistical metrics. Since the analyses were performed using established bioinformatics portals (e.g., UALCAN, TIMER, bc-GenExMiner), the results presented are based on pre-processed datasets, and only statistically significant outputs (typically with p-values < 0.05) were extracted and reported. These platforms apply internal statistical tests and thresholds, and we have now clarified this in the revised manuscript.

5. Expand figure legends to include the context, sample sizes, data source, and interpretation of key trends. Use arrows or color-coded annotations to highlight significant findings.

Response: We thank the reviewer for this suggestion. We have already addressed this comment while responding to other reviewers by expanding the figure legends.

6. The translational potential (e.g., application to TNBC, resistance cases, or prognosis) is mentioned but not convincingly argued. Clearly define how co-targeting TGFβ-1 and CXCR4 could be integrated into existing treatment paradigms, and whether this is more suitable for specific molecular subtypes.

Response: We thank the reviewer for this important point. We have already addressed this comment in the revised manuscript under the Discussion section, while responding to Reviewer 1 and Reviewer 2. Specifically, we elaborated on the translational potential of co-targeting TGFβ-1 and CXCR4, particularly in TNBC and therapy-resistant cases, and discussed how this strategy could be integrated into current treatment paradigms based on molecular subtype-specific vulnerabilities.

7. The rationale for choosing D4476 and AMD3100 is weak, and potential off-target effects are not adequately addressed. Please provide literature-based justification for drug selection and discuss anticipated toxicological risks in normal tissues expressing TGFβ-1 and CXCR4.

Response: We thank the reviewer for this observation. We have already discussed the rationale for selecting D4476 and AMD3100, along with the potential off-target effects and toxicological risks, in the revised manuscript under the Discussion section, while addressing related comments from Reviewers 1 and 2

Minor Issues:

1. A native-level language editor should review English grammar, syntax, and phrasing. Numerous typographical and stylistic issues remain.

Response: We appreciate the reviewer’s feedback. The revised manuscript has been thoroughly reviewed and edited by a native-level English language editor to correct all remaining grammatical, syntactic, typographical, and stylistic issues.

2. Use gene/protein nomenclature consistently (e.g., “CXCR4” vs “Cxcr4”).

Response: We thank the reviewer for pointing this out. In the revised manuscript, we have ensured consistent use of gene and protein nomenclature throughout, following standard conventions (e.g., human genes in uppercase italics, such as CXCR4, and proteins in uppercase regular font, such as CXCR4). All gene and protein names have been reviewed and corrected accordingly.

---

## [Decision Letter · Decision Letter 2]

8 Aug 2025

Unravelling the Therapeutic Potential of Dual TGFβ-1 and CXCR4 Inhibition in Breast Cancer Using Computational Strategies

PONE-D-25-01591R2

Dear Dr. Mir,

We’re pleased to inform you that your manuscript has been judged scientifically suitable for publication and will be formally accepted for publication once it meets all outstanding technical requirements.

Kind regards,

Antimo Migliaccio, M.D.

Academic Editor

PLOS ONE

Additional Editor Comments (optional):

Reviewers' comments:

Reviewer's Responses to Questions

**Comments to the Author**

Reviewer #3: All comments have been addressed

Reviewer #4: All comments have been addressed

2. Is the manuscript technically sound, and do the data support the conclusions?

Reviewer #3: Yes

Reviewer #4: Yes

3. Has the statistical analysis been performed appropriately and rigorously?

Reviewer #3: Yes

Reviewer #4: Yes

4. Have the authors made all data underlying the findings in their manuscript fully available?

Reviewer #3: Yes

Reviewer #4: Yes

5. Is the manuscript presented in an intelligible fashion and written in standard English?

Reviewer #3: Yes

Reviewer #4: Yes

Reviewer #3: (No Response)

Reviewer #4: The revised manuscript demonstrates a thoughtful and constructive response to the previous concerns. The authors have taken significant steps to improve the clarity, depth, and translational framing of their computational analysis.

---

## [Editor Report · Acceptance letter]

PONE-D-25-01591R2

PLOS ONE

Dear Dr. Mir,

I'm pleased to inform you that your manuscript has been deemed suitable for publication in PLOS ONE. Congratulations! Your manuscript is now being handed over to our production team.

Kind regards,

on behalf of

Dr. Antimo Migliaccio

Academic Editor

PLOS ONE